



# SGD-SM: Generating Seamless Global Daily AMSR2 Soil Moisture Long-term Productions (2013-2019)

Qiang Zhang[1], Qiangqiang Yuan[2, 3] *, Jie Li[2], Yuan Wang[2], Fujun Sun[4], and Liangpei Zhang[1]

[1]State Key Laboratory of Information Engineering, Survey Mapping and Remote Sensing, Wuhan University, China
[2]School of Geodesy and Geomatics, Wuhan University, China
[3]Collaborative Innovation Center of Geospatial Technology, Wuhan University, China
[4]Beijing Electro-mechanical Engineering Institute, Beijing, China

**Correspondence:** Qiangqiang Yuan (yqiang86@gmail.com)

**Abstract.** High quality and long-term soil moisture productions are significant for hydrologic monitoring and agricultural management. However, the acquired daily soil moisture productions are incomplete in global land (just about 30%~80% coverage ratio), due to the satellite orbit coverage and the limitations of soil moisture retrieving algorithms. To solve this inevitable problem, we develop a novel 3D spatio-temporal partial convolutional neural network (CNN) for Advanced Microwave Scanning

Radiometer 2 (AMSR2) soil moisture productions gap-filling. Through the proposed framework, we generate the seamless global daily (SGD) AMSR2 soil moisture long-term productions from 2013 to 2019. To further validate the effectiveness of these productions, three verification ways are employed as follow: 1) In-situ validation; 2) Time-series validation; And 3) simulated missing regions validation. Results show that the seamless global daily soil moisture productions have reliable cooperativity with the selected in-situ values. The evaluation indexes of the reconstructed (original) dataset are R: 0.683 (0.687), RMSE:

0.099 $m^3/m^3$ (0.095 $m^3/m^3$), and MAE: 0.081 $m^3/m^3$ (0.078 $m^3/m^3$), respectively. Temporal consistency of the reconstructed daily soil moisture productions is ensured with the original time-series distribution of valid values. Besides, the spatial continuity of the reconstructed regions is also accorded with the context information (R: 0.963~0.974, RMSE: 0.065~0.073 $m^3/m^3$, and MAE: 0.044~0.052 $m^3/m^3$). More details of this work are released at **https://qzhang95.github.io/Projects/Global-Daily-Seamless-AMSR2/**. This dataset can be downloaded at **https://zenodo.org/record/3960425** (Zhang et al., 2020. DOI:

10.5281/zenodo.3960425).

## 1 Introduction

Surface soil moisture is a crucial earth land characteristic in describing of hydrologic cycle system (Wigneron et al., 2003; Lievens et al., 2015). It can be applied for monitoring the droughts and floods phenomena of agriculture (Samaniego et al., 2018) and geologic hazards (Long et al., 2014). To obtain the global and high-frequency soil moisture products, many active

or passive satellite sensors have been launched such as AMSR-E, AMSR2, ESA CCI, SMAP, SMOS and so on (McColl et al., 2017; Ma et al., 2019). Nevertheless, the acquired daily soil moisture productions are always incomplete in global land (about 30%~80% missing ratio in AMSR2), because of the satellite orbit coverage and the limitations of soil moisture retrieving algorithms (Cho et al., 2017; Long et al., 2019). Especially in the regions close to the equator, or in the permafrost region,



the soil moisture data missing degree is more serious (Zeng et al., 2015; Santi et al., 2018). This phenomenon greatly disturbs

subsequent soil moisture applications, especially for the consecutive daily temporal analysis and global spatial-distribution comparisons (Colliander et al., 2017).

To reduce this negative effect, most existing works employed the strategy of multi-temporal soil moisture data selecting, multi-temporal soil moisture data averaging, or multi-temporal soil moisture data synthesizing to deal with this inevitable problem. Detailed descriptions and analysis of these three strategies are presented as follows: (Bitar et al., 2017)

1) **Multi-temporal soil moisture data selecting**: Criterion of this strategy denotes to selecting the highest coverage regions in single date from multi-temporal soil moisture productions (Wang and Qu, 2009). However, this assumption can only deal with local regions, and not applicable for global regions. The main reason is that almost all the global daily soil moisture productions suffer from the defect of satellite orbit coverage missing and retrieving algorithm failure. Besides, multi-temporal soil moisture data selecting strategy greatly reduce the data utilization, and is also not able to qualified for dense time-series

analysis on daily temporal resolution (Purdy et al., 2018).

2) **Multi-temporal soil moisture data averaging**: This strategy is commonly used for most soil moisture study or applications. The incomplete soil moisture productions are overall averaged as the monthly/quarterly/yearly results to generate the complete productions (Jalilvand et al., 2019). For most applications and spatial analysis, this operation can effectively improve the spatial soil moisture coverage (Zhao et al., 2020). However, it distinctly sacrifices the high-frequency temporal resolution

as low-frequency temporal resolution, which also severely reduces the data utilization. Besides, it ignores the unique spatial-distribution of single day and loses the dense time-series changing information. In other word, the monthly/quarterly/yearly soil moisture data averaging operations damage the initial information on both spatial and temporal dimension.

3) **Multi-temporal soil moisture data synthesizing**: Different from soil moisture data selecting and averaging, this strategy employs the time-series daily soil moisture data and selects the best observed value from corresponding time-series pixels.

This strategy can produce synthesizing result through best single-point, while it ignores the spatial local correlation and exists incontinuous and inconsistent effects in local regions. In addition, it also sacrifices high temporal resolution just as multi-temporal data averaging strategy (Peng et al., 2017).

To overcome above-mentioned limitations, some missing values reconstruction methods have been developed especially on multi-temporal images thick cloud removal and deadline gap-filling (Zhang et al., 2020a). For example, Zhu et al. (2011)

proposed the multi-temporal neighboring homologous value padding method for thick cloud removal. Chen et al. (2011) presented an effective interpolating algorithm for recovering the invalid regions in Landsat images. Zhang et al. (2018a) built an integrative spatio-temporal-spectral network for missing data reconstruction in multiple tasks. And for the soil moisture productions gap-filling, some methods have also been proposed to address this issue. Wang et al. (2012) presented a penalized least square regression-based approach for global satellite soil moisture gap filling observation. Fang et al. (2017) introduced a

long short-term memory network to generate spatial complete overlay SMAP in U.S. Long et al. (2019) fused multi-resolution soil moisture productions, which can produce daily fine-resolution data in local regions. Llamas et al. (2020) used geostatistical techniques and multiple regression strategy to get spatial complete results of satellite-derived productions. Overall, there are few works for soil moisture productions reconstructing on global and daily scale.





In spatial dimension, the invalid land areas and adjacent valid land areas exist the context consistency and spatial correlation on daily soil moisture productions (Cui et al., 2016). In temporal dimension, daily time-series changing curve of the same point natively appears with the continuous and smooth peculiarities (Chan et al., 2018). Therefore, how about simultaneously extracting both spatial and temporal features for seamless soil moisture data reconstructing? Accordingly, some intuitions of this work are spontaneously raised below:

a) How to turn waste into wealth to enhance the soil moisture data utilization, and develop a new strategy to solve the incomplete spatio-temporal dilemma of satellite soil moisture productions;

b) How to obtain the global gap-filling productions and simultaneously remain the daily temporal resolution, and ensure the reliable precision of these productions;

c) How to utilize current deep learning theory, and effectively extract the spatio-temporal information to generate global long-term soil moisture productions.

From these perspectives, a novel 3-D spatio-temporal partial convolutional neural network is proposed for AMSR2 soil moisture productions gap-filling. By means of the proposed method, we can effectively break through the above-mentioned limitations. And finally, this work generates the seamless global daily AMSR2 soil moisture long-term productions from 2013 to 2019. The main innovations are summarized as below:

1) We develop a deep 3D spatio-temporal partial reconstruing model, which can take both the spatial and temporal information into consideration. Aiming at the invalid or coastline region boundary, the 3D partial CNN and global-local loss function are presented for better extracting the valid region features and ignoring the invalid regions through both soil moisture data and mask information.

2) A seamless global daily (SGD) AMSR2 soil moisture long-term (2013-2019) dataset is generated through the proposed model. The dataset includes the original and reconstructing soil moisture data. And this SGD productions could be directly downloaded at **https://qzhang95.github.io/Projects/Global-Daily-Seamless-AMSR2/** (Zhang et al., 2020).

3) Three verification strategies are employed to testify the precision of our SGD soil moisture dataset as follows: in-situ validation; time-series validation; and simulated missing regions validation. Evaluating indexes demonstrate that the seamless global daily AMSR2 soil moisture dataset shows high accuracy, reliability, and robustness.

The schema of this work is listed below. Section 2 describes the study ASMR2 soil moisture productions and in-situ soil moisture network data. Section 3 presents the methodology for generating the seamless global daily AMSR2 soil moisture productions. Section 4 gives the experimental results and related validation results. Besides, some comparisons between time-series averaging method and proposed method are discussed in Section 5. And at last, Section 6 makes the conclusions of this study.



## 2 Data description

### 2.1 AMSR2 soil moisture productions

In this work, we focus on dealing with AMSR2 soil moisture productions products. This sensor was onboard on the Global Change Observation Mission 1-Water (GCOM-W1) satellite, launched in May 2012 (Kim et al., 2015). The released datasets include three passive microwave band frequencies: 6.9 GHz (C1 band), 7.3 GHz (C2 band, new frequency compared with AMSR-E), and 10.7 GHz (X band). It can observe the global land two times within a day (Wu et al., 2016): ascending (day-
time) and descending (night-time) orbits. The primary spatial resolution of this datasets denotes $0.25°$ global grids. And the AMSR2 soil moisture retrieval algorithms include Land Parameter Retrieval Model (LPRM) and Japan Aerospace Exploration Agency (JAXA) (Du et al., 2017; Kim et al., 2018). Besides, the uncertainty of soil moisture for each band were also given in AMSR2 productions.

     In our study, we choose LPRM AMSR2 descending level 3 (L3) global daily $0.25°$ soil moisture productions as the research
data. This dataset could be obtained from https://hydro1.gesdisc.eosdis.nasa.gov/. For instance, the original AMSR2 $0.25°$ soil moisture data obtained in April 2, 2019 is displayed in Fig. 1. Due to the satellite orbit coverage and the limitations of soil moisture retrieving algorithms in tundra areas (Muzalevskiy et al., 2020), the acquired AMSR2 daily soil moisture productions are always incomplete in global land (about 30%∼80% invalid ratio, excluding Antarctica and most of Greenland), as shown in Fig. 1. The daily global land coverage ratio of AMSR2 soil moisture data in 2019 is listed in Fig. 2. Distinctly, the global
land coverage ratio is low in wintertime, and high in summertime. The mean global coverage ratio of 2019 is just about 56.5% in AMSR2 soil moisture daily productions. Apparently, these incomplete soil moisture data cannot be directly applied for subsequent spatial and time-series analysis, as mentioned in previous Section 1.

### 2.2 International Soil Moisture Network in-situ data

     The International Soil Moisture Network (ISMN) was established from 2009 to now (Dorigo et al., 2011), which was
employed to provide the correction/validation schemes for remote sensing satellite-based soil moisture retrieval. ISMN includes the globally distributed in-situ soil moisture sites supported by the earth observation of the European Space Agency (ESA) and many voluntary contributions of researchers and organizations from all over the world (Dorigo et al., 2012; Dorigo et al., 2013).

     The ISMN in-situ surface soil moisture values could be acquired through https://ismn.geo.tuwien.ac.at. In our experiments, we selected part in-situ soil moisture sites of ISMN as ground truth values (Zhang et al., 2017), to testify the precision and
credibility of the reconstructing datasets in Section 4.2. It should be noted that the time range is restrained from 2013.1.1 to 2019.12.31. These in-situ sites are matched with the corresponding daily soil moisture productions and related locations. Two neighboring in-situ hourly values are then averaged as the ultimate result of current date (Dong et al., 2020).



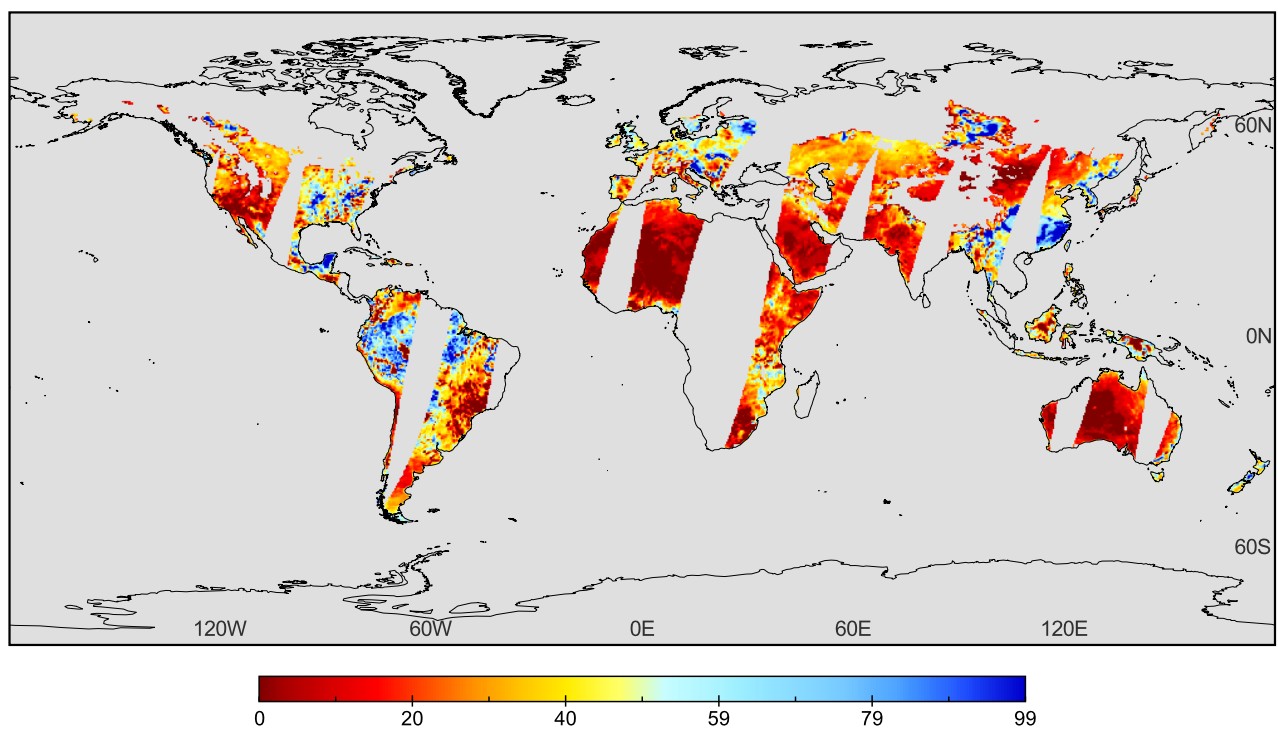

**Figure 1.** Original global AMSR2 0.25° soil moisture data (obtained in April 2, 2019)

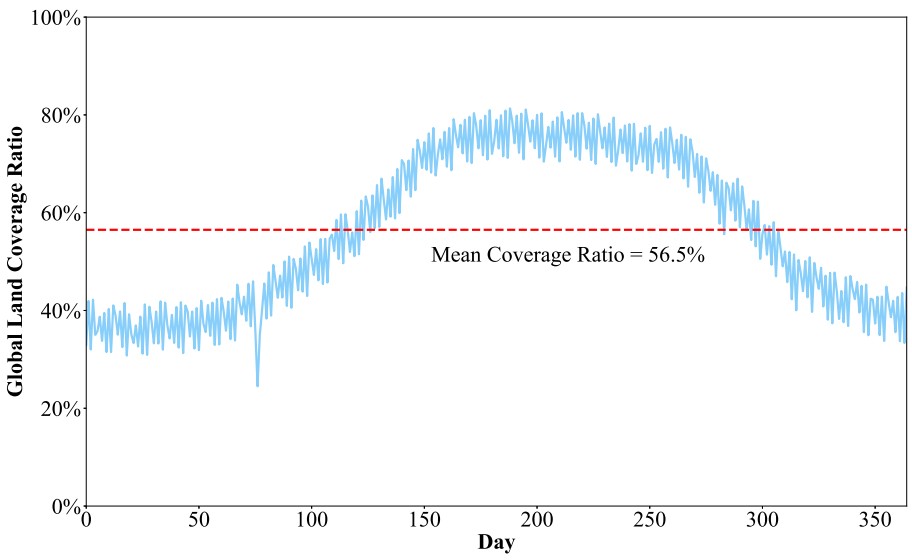

**Figure 2.** The daily global land coverage ratio of AMSR2 soil moisture productions in 2019



## 3 Methodology

The flowchart of the presented framework is depicted in Fig. 3. Firstly, we designate the processing daily soil moisture data in
date $T$, and simultaneously select its adjacent time-series data before and after four days (date $T$-4 to $T$+4). The corresponding
land masks of these daily soil moisture data are generated through the invalid pixel marking. The overall structure could be
decomposed as two stages: the training procedure and testing procedure, as described in Fig. 3.

In the training procedure, these spatio-temporal soil moisture data and land mask patch groups are imported as the training
data of the presented spatio-temporal 3-D reconstructing model through patch selecting and mask simulating. Then in the
testing procedure, seamless global daily reconstructing soil moisture data is outputted through the loss convergent model.
Subsequently, the next processing daily soil moisture data is designated and repeat above-mentioned steps, until all the daily
data are serially reconstructed in order. Details of the reconstructing model and network are described below.

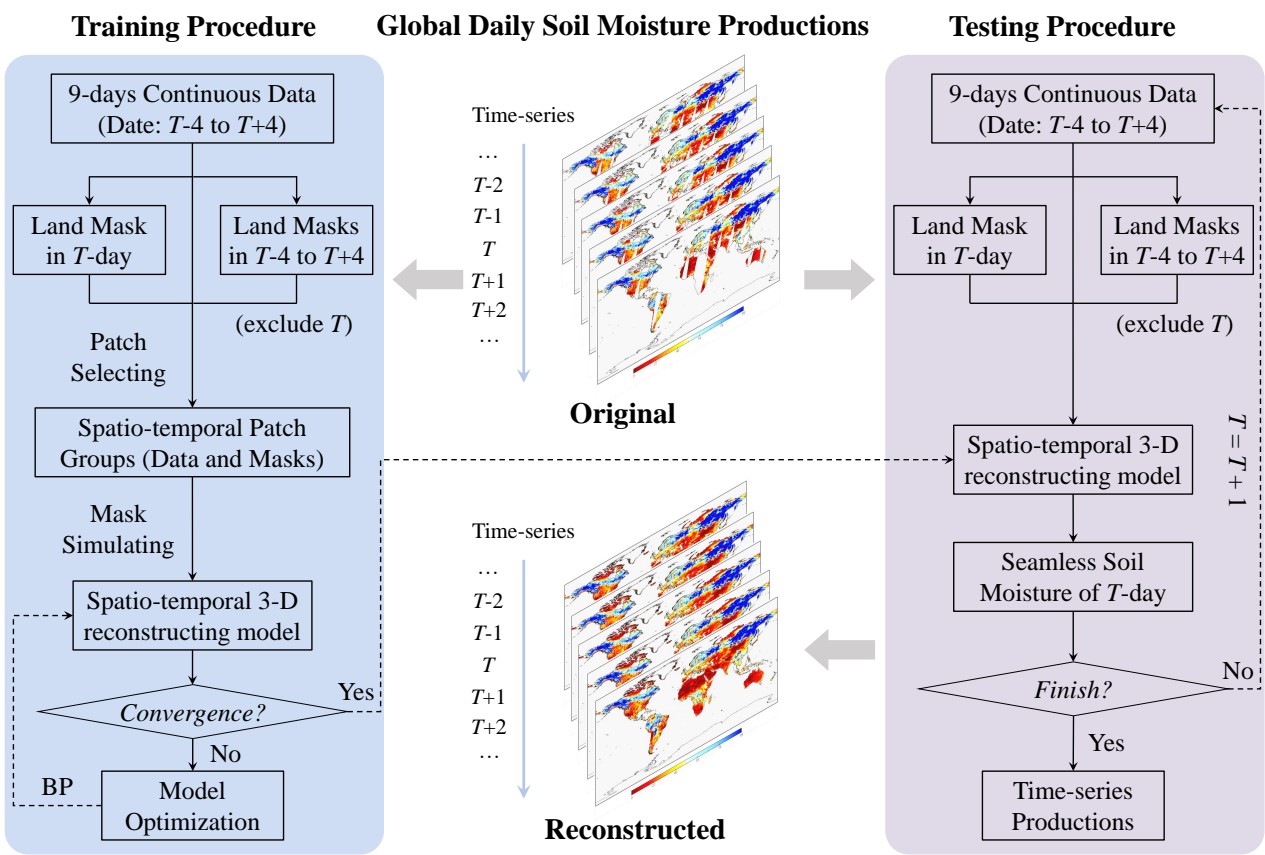

**Figure 3.** Flowchart of the presented framework





### 3.1 Spatio-temporal 3-D reconstructing model

The spatio-temporal 3-D soil moisture reconstructing model is displayed in Fig. 4. After assigning the original soil moisture
data in date $T$, time-series soil moisture data and corresponding masks in date $T$-4 to $T$+4 are simultaneously imported as the
3D-tensor inputs of the presented deep reconstructing model in Fig. 4. In spatial dimension, missing and non-missing areas
exist the context consistency in daily soil moisture data. In temporal dimension, the daily time-series changing curve of the
same point natively appears with the continuous and smooth peculiarities. Therefore, the 3D CNN is employed to process
the spatio-temporal 3-D soil moisture data in this model. Through this way, we can jointly utilize both spatial and temporal
information of these time-series soil moisture productions. Further, it can better richly exploit the deep spatio- temporal feature
for data reconstructing and model optimization. The structure and details are depicted in Fig. 4.

This network includes 11 layers (3D partial CNN unit and ReLU (Rectified Linear Unit)) in Fig. 4. The size of 3D filters is
all set as 3×3×3. Number of feature maps before ten layers is fixed as 90, and the channel of feature map in the final layer is
exported as 1. It should be noted that after finishing each partial 3D-CNN layer, we must update all the new masks for next
layer. The mask updating operation is defined in Section 3.2. In terms of the model training and optimization, three steps: patch
selecting, mask simulating, and back propagation are performed in Section 3.3. Besides, we take the global loss and local loss
into consideration for network optimization. As described in Fig. 3, this deep reconstructing model need to be learned with
large training label samples, before the testing procedure for outputting global seamless daily soil moisture productions. The
global land mask and the mask in current date $T$ are also employed for the global loss and local loss in Fig. 4. Descriptions of
partial 3D-CNN and model optimization are demonstrated in Section 3.2 and Section 3.3, respectively.

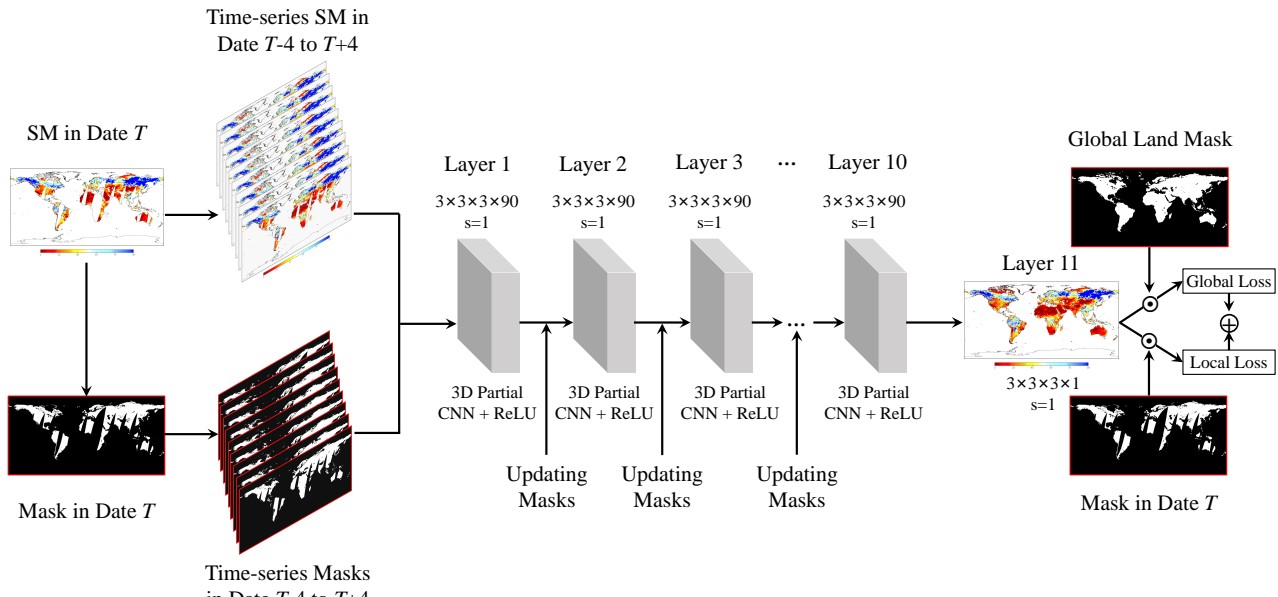

**Figure 4.** Spatio-temporal 3-D soil moisture reconstructing model



## 3.2 Partial convolutional neural network

Deep convolution neural network has been widely applied for nature image reconstructing (Liu et al., 2018a; Yeh et al., 2017; Liu et al., 2019) and satellite imagery recovering (Yuan et al., 2019; Zhang et al., 2019; Zhang et al., 2020b). Nevertheless, it should be highlighted that the valid and invalid pixels simultaneously exist especially around the coast regions and gap regions

(Pathak et al., 2016). The common CNN ignores the location information of invalid or valid pixels in soil moisture data, which cannot eliminate the invalid information (Liu et al., 2018b). Therefore, to solve this negative effect, we develop the partial 3D-CNN to ignore the invalid information in the proposed reconstructing model.

Before introducing the partial convolution, the operation of common convolution in most deep learning framework can be defined as below:

$$x = \mathbf{W}^T \mathbf{X} + b \tag{1}$$

where $\mathbf{X}$ denotes the inputted tensor data. $\mathbf{W}$ and $b$ are the weight and bias parameters, respectively. Different from the common convolution, the mask information $\mathbf{M}$ of the corresponding soil moisture data is introduced into the partial convolution:

$$x' = \begin{cases} \mathbf{W}^T(\mathbf{X}_{(w,h,t)} \odot \mathbf{M}_{(w,h,t)}) \dfrac{\left\|\mathbf{1}_{(w,h,t)}\right\|_1}{\left\|\mathbf{M}_{(w,h,t)}\right\|_1} + b, & \left\|\mathbf{M}_{(w,h,t)}\right\|_1 \neq 0 \\ 0, & otherwise \end{cases} \tag{2}$$

where $\odot$ stands for the pixel-wise multiplication. $\mathbf{1}$ denotes the identical dimension tensor with mask $\mathbf{M}$, whose elements are

all value 1. Obviously, the partial convolutional output $x'$ is only decided by the valid soil moisture pixels of input $\mathbf{X}$, rather than the invalid soil moisture pixels. Through the mask $\mathbf{M}$, we can effectively exclude the interference information of invalid soil moisture pixels such as marine regions and gap regions. Then the scaling divisor in Eq. (2) further adjusts for the variational number of valid soil moisture pixels.

After finishing each partial convolution layer, all the masks need to updated through the following rule: If the partial con-

volution can generate at least one valid value of the output result, then we mark this location as valid value in the new masks. This updating operation is demonstrated as below:

$$m'_{(w,h,t)} = \begin{cases} Land_{(w,h)} \cdot 1, & \left\|\mathbf{M}_{(w,h,t)}\right\|_1 \neq 0 \\ 0, & otherwise \end{cases} \tag{3}$$

where $Land_{(w,h)}$ is the global land mask in location $(w,h)$ of the global soil moisture production. This global land mask covers six continents and excludes Antarctica and most of Greenland.





### 3.3 Model training and optimization

As shown in Fig. 3, the training procedure need generate large numbers of training samples for learn the proposed spatio-temporal 3-D reconstructing model in Fig. 4. Different from the testing procedure, the training procedure additionally contains the patch selecting, mask simulating, and back propagation (BP) steps. These three steps are significant for model training and optimization. The purpose of patch selecting and mask simulating step in Fig. 3 is to establish the label (complete)-data (incomplete) training samples in the deep learning framework. Besides, the significance of BP step in Fig. 3 is to optimize the reconstructing network in Fig. 4 and acquire the loss convergence model for testing use.

In the patch selecting step, we traverse the global regions in date $T$ to select the complete soil moisture patch label, whose local land regions are undamaged. It should be noted the rest incomplete patches in date $T$ are excluded because they cannot participate in the supervised learning. The corresponding time-series soil moisture patches of this selected patch between date $T$-4 to $T$+4, is set as the spatio-temporal 3D data patch groups. And their corresponding masks between date $T$-4 to $T$+4 is set as the spatio-temporal 3D mask patch groups. After traversing the original global daily AMSR2 soil moisture productions from 2013 to 2019, we finally establish the spatio-temporal data and mask patch groups with the number of 276488 patches. The soil moisture patch size is fixed as 40×40 for patch selecting.

In the mask simulating step, 10000 patch masks of the size 40×40 are chosen from the global AMSR2 soil moisture masks from 2013 to 2019. The missing ratio range of these masks is set as [0.3, 0.7]. Then these patch masks are randomly selected for label patches use within the spatio-temporal data and mask patch groups. The complete patch in date $T$ (label) is simulated as the incomplete patch (data) through the above mask. And the original corresponding mask of this patch needs also to be replaced. After traversing and building the label-data 3D spatio-temporal patch groups, this dataset is set as the training samples for the usage of reconstructing network in Fig. 3.

In the back propagation step, we need a loss function to iteratively optimal the learning parameters of the deep reconstructing network. This operation follows the chain rule in model optimizing. The Euclidean loss function is employed in most data reconstruction or regression issues based on deep learning, such as satellite productions downscaling (Fang et al., 2020) and retrieving (Lee et al., 2019). Nevertheless, Euclidean loss function only pays attention to the holistic information bias for network optimization. It ignores the soil moisture particularity of the local areas, especially in local coastal, mountain, and hinterland regions. However, this particularity is extremely significant for invalid regions gap-filling, because of the spatial heterogeneity in soil moisture productions. Therefore, to take both the global consistency and local soil moisture particularity into consideration, the global land mask and current mask in date $T$ are both employed after the final layer as shown in Fig. 3. Through this way, we can simultaneously ensure the global consistency and distinguish the local particularity. Further, the reconstructing network presents the local and global loss as below:

$$\zeta_{local} = \|(1 - \mathbf{M}_T) \odot (\mathbf{SM}_{rec} - \mathbf{SM}_{ori})\|_2^2 \tag{4}$$





$$\zeta_{global} = \|\mathbf{M}_G \odot (\mathbf{SM}_{rec} - \mathbf{SM}_{ori})\|_2^2 \tag{5}$$

where $\mathbf{M}_T$ stands for current mask patch in date $T$. $\mathbf{M}_G$ represents the corresponding global land mask patch. $\mathbf{SM}_{rec}$ and $\mathbf{SM}_{ori}$ denote the reconstructed soil moisture patch and original seamless soil moisture patch, respectively. The unified loss function of the reconstructing network combines $\zeta_{local}$ and $\zeta_{global}$ as below:

$$\zeta(\mathbf{\Theta}) = \zeta_{local} + \eta \cdot \zeta_{global} \tag{6}$$

where $\mathbf{\Theta}$ refers to the learnable arguments for each layer of the deep reconstructing model. $\eta$ denotes the balancing factor to adjust the $\zeta_{local}$ and $\zeta_{global}$. In this work, we fixed this factor as 0.1 during the training procedure.

After building up this unified loss function, the presented reconstructing model employs Adam algorithm as the gradient descent strategy. The number of batch size in this model is fixed as 128 for network training. The total epochs and initial learning
rate are determined as 300 and 0.001, respectively. Starting every 30 epochs, the learning rate is degraded through decay coefficient 0.5 (Zhang et al., 2018b). Besides, the proposed network uses the Pytorch (https://pytorch.org/) platform to test and train the deep soil moisture reconstructing model. The software environment is listed as follows: Python 3.7.4 language, Windows 10 operating system, and PyCharm 2019 integrated development environment (IDE). The final soil moisture productions are exported as netCDF4 hierarchical data format, which both contains the original and reconstructed soil moisture data.

**4   Experimental results and validation**

In this section, we provide the experimental results and related validation results to testify the availability of the presented framework. Through this framework, we finally generate the seamless global daily AMSR2 soil moisture long-term productions from 2013.1.1 to 2019.12.31. The daily soil moisture products are saved as NetCDF4 format. It should be highlighted that this dataset can be directly downloaded at **https://qzhang95.github.io/Projects/Global-Daily-Seamless-AMSR2/** for free-
use. Besides, more details and experimental results of this dataset are displayed in this website because of the paper layout limitations. And the related codes of this dataset are also available at **https://github.com/qzhang95/SGD-SM**.

We firstly give two sample seamless reconstructing results of global time-series soil moisture productions. The original and reconstructed results are both given for comparisons. Later, to further validate the effectiveness of these productions, three verification ways are respectively employed as follows:
1) In-situ validation.

2) Time-series validation.

3) Simulated missing regions validation.





### 4.1 Experimental results

As displayed in Fig. 5 (a)-(h) and Fig. 6 (a)-(h), original and reconstructing global daily time-series AMSR2 soil moisture
productions between 2019.6.1 to 6.4 and 2016.10.1 to 10.4 are given as the sample results, respectively. The left column lists
the original incomplete soil moisture results, and the right column lists the corresponding complete soil moisture results after
reconstructed by the proposed method in Fig. 5 and Fig. 6. We ignore the regions of Antarctica and most of Greenland, because
the satellite soil moisture data within these regions behaves perennially missing.

From the spatial dimension, the reconstructing global soil moisture productions are consistent between invalid regions and
their adjacent valid regions in Fig. 5 and Fig. 6. Especially around the high-value areas and low-value areas, the context
information consecutive without obvious reconstructing boundary effects such as in Africa, Australia, and Europe in Fig. 5 and
Fig. 6.

From the temporal dimension, although the incomplete time-series daily results are highly similar and correlative, there
are still some variations and differences between each other. The proposed method performs well on consistent temporal
information preserving and specific temporal information predicting in Fig. 5 and Fig. 6.



(a) Original SM in 2019.6.1

(b) Reconstructing SM in 2019.6.1

(c) Original SM in 2019.6.2

(d) Reconstructing SM in 2019.6.2

(e) Original SM in 2019.6.3

(f) Reconstructing SM in 2019.6.3

(g) Original SM in 2019.6.4

(h) Reconstructing SM in 2019.6.4

**Figure 5.** Original/reconstructing global daily SM results between 2019.6.1 to 6.4





(a) Original SM in 2016.10.1

(b) Reconstructing SM in 2016.10.1

(c) Original SM in 2016.10.2

(d) Reconstructing SM in 2016.10.2

(e) Original SM in 2016.10.3

(f) Reconstructing SM in 2016.10.3

(g) Original SM in 2016.10.4

(h) Reconstructing SM in 2016.10.4

**Figure 6.** Original/reconstructing global daily SM results between 2016.10.1 to 10.4



## 4.2 In-situ validation

In-situ shallow-depth soil moisture sites can be employed as the ground-truth to validate the reconstructing satellite soil moisture productions. We select 125 soil moisture stations (0 10cm) through ISMN between 2013.1.1 to 2019.12.31. Nine soil moisture in-situ sites and the corresponding reconstructing data within invalid regions are then contrast as the scatter plots in
Fig. 7 (a)-(i), respectively. The horizontal axis stands for in-situ soil moisture value. Meanwhile the vertical axis represents reconstructing soil moisture value. It should be highlighted that due to the lacks of part recorded data between 2013 to 2019, most in-situ values are incompleteness with different point numbers. As shown in Fig. 7 (a)-(i), the correlation coefficients (R) indexes are distributed between 0.679 to 0.754. The root-mean-square error (RMSE) and mean absolute error (MAE) indexes are distributed from 0.026 m$^3$/m$^3$ to 0.134 m$^3$/m$^3$ and from 0.021 m$^3$/m$^3$ to 0.107 m$^3$/m$^3$, respectively.

**Figure 7.** Scatters of the in-situ/reconstructed soil moisture values within selected stations





In additions, we compare the reconstructed with original AMSR2 soil moisture productions through the selected 125 in-situ sites, as listed in Table 1. The averaged R, RMSE, and MAE of the original and reconstructed soil moisture productions are 0.683 (0.687), 0.099 m$^3$/m$^3$ (0.095 m$^3$/m$^3$), and 0.081 m$^3$/m$^3$ (0.078 m$^3$/m$^3$), respectively. Overall, the accuracy of reconstructed soil moisture productions is generally accorded with the original productions. The differences of these indexes R, RMSE and MAE are minor between the original and reconstructed soil moisture results in Table 1. To some degree, this
validation ensures the reliability and availability of the proposed seamless global daily AMSR2 soil moisture productions.

**Table 1.** Comparisons between original and reconstructed soil moisture productions

| Soil Moisture Productions | Evaluation Index | | |
|---|---|---|---|
| | R | RMSE (m$^3$/m$^3$) | MAE (m$^3$/m$^3$) |
| Original | 0.687 | 0.095 | 0.078 |
| Reconstructed | 0.683 | 0.099 | 0.081 |

### 4.3   Time-series validation

To further validate the reconstructed soil moisture results, time-series variations of both original and reconstructed results are stacked in six regions around the six continents: Africa (0.375°N, 36.875°E), Europe (49.375°N, 35.125°E), Asia (38.125°N, 117.375°E), North America (39.875°N, 106.125°W), South America (15.125°S, 52.625°W), Australia (30.125°S,
150.375°E), respectively. As described in Fig. 8(a)-(f), the horizontal axis stands for daily time-series date between 2013 to 2019. The vertical axis represents the soil moisture value. The blue points refer to the original valid soil moisture daily results, and the red forks stands for the reconstructed invalid soil moisture daily results in Fig. 8.

As depicted in Fig. 8(a)-(f), most of the soil moisture time-series scatters can obviously reveal the annual periodic variations. The reconstructed soil moisture results generally behave fine temporal consistency with the original soil moisture results in
different areas. Related low soil moisture values mostly existed in the droughty season of winter with the frozen lands such as in Fig. 8(d). Related high soil moisture values mainly generated in the moist season of summer with more rainy days, especially in Fig. 8 (b), (d) and (f).

Overall, compared with the whole original variation tendency between 2013 to 2019, the generated seamless global daily AMSR2 soil moisture long-term productions can steadily reflect the temporal consistency and variation. It is significant for
time-series applications and analysis. This daily time-series validation also demonstrates the robustness of the presented method and the availability of the established seamless global daily productions.

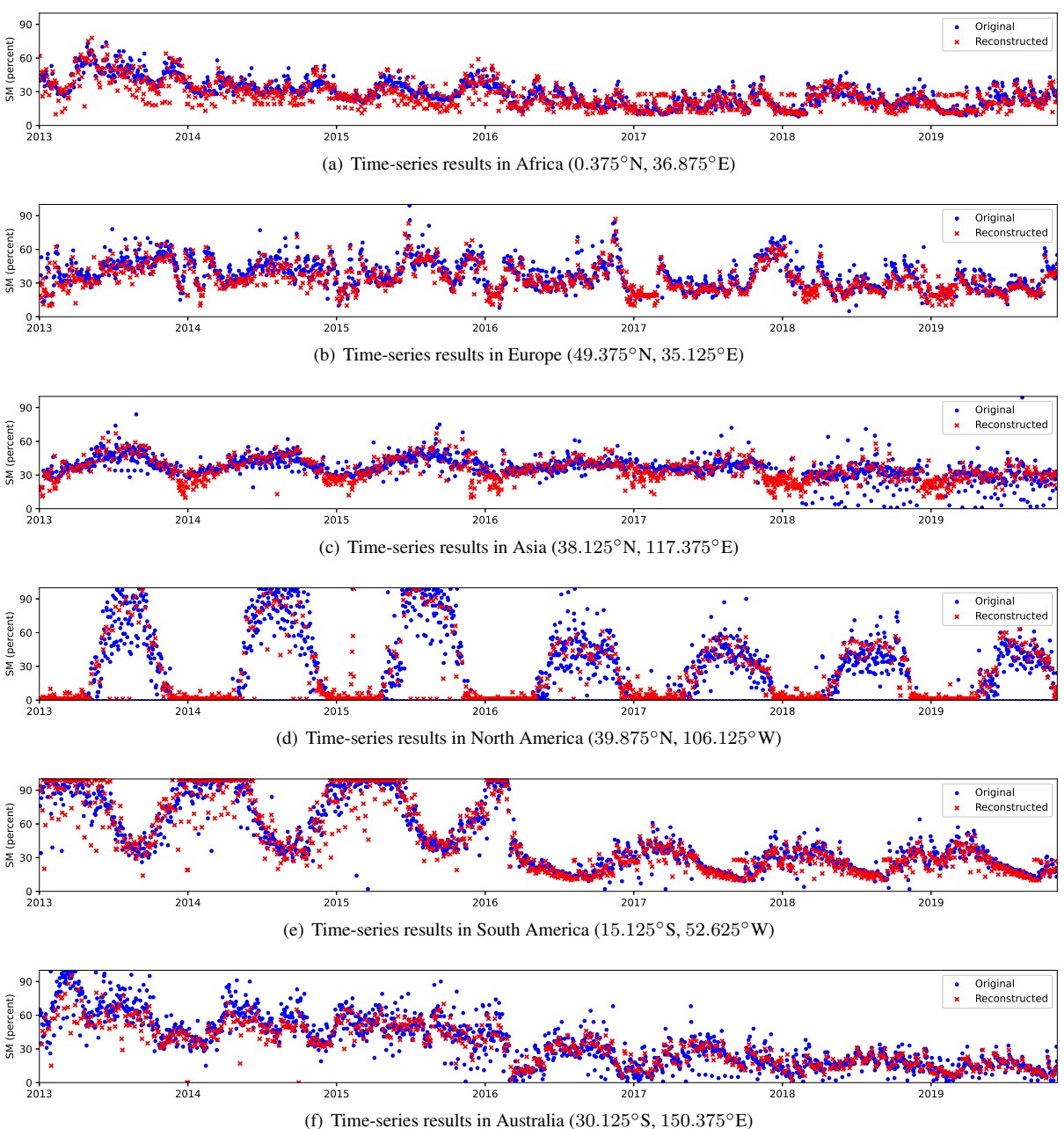

**Figure 8.** Original/reconstructed time-series results in selected regions





## 4.4 Simulated missing regions validation

In addition to the time-series consistency in Section 4.3, the spatial continuity is also important for the reconstructed seamless soil moisture productions. Therefore, to further testify this key point, we carry out the simulated missing regions validation in this subsection. Based on the original soil moisture productions, six simulated square missing regions are performed in six continents, respectively. We select four dates of the long-term soil moisture productions: 2013.7.25, 2015.7.25, 2017.7.25, and 2019.7.25 as the simulated objects. For example, original and reconstructed results with simulated missing regions in 2019.7.25 are depicted in Fig. 9(a) and (b), respectively. The simulated missing regions can be clearly observed in Fig. 9(a) around the six continents. Besides, detailed original and reconstructed spatial information of four simulated patches in 2015.7.25 are displayed in Fig. 10. Table 2 gives the evaluation index (R, RMSE, MAE) of the simulated patches between 2013 to 2019. Then the original-reconstructed scatters of simulated regions in 2013, 2015, 2017, and 2019.7.25 are listed in Fig. 11(a)-(b), respectively.

As shown in Fig. 9 (a) and (b), the reconstructed invalid regions are consecutive between the original valid regions. And in the simulated missing regions, the spatial texture information is also continuous without obvious boundary reconstructing effects in Fig. 9(b). To better distinguish the spatial details of reconstructed soil moisture, we select four enlarged patches in simulated regions in Fig. 10. It can be clearly observed that the reconstructed patches perform the high consistency with the original patches, as displayed in Fig. 10.

Besides, the reconstructed soil moisture patches in simulated missing regions behave high reconstructing accuracy, whose R values are distributed between 0.963 to 0.974 in Table 2 and Fig. 11(a)-(d). RMSE and MAE values also perform well on 0.065 to 0.073 $m^3/m^3$ and 0.044 to 0.052 $m^3/m^3$ in Table 2 and Fig. 11(a)-(d), respectively. Overall, this simulated missing regions validation manifests the reconstructing ability of spatial information continuity.

**Table 2.** Evaluation indexes of the simulated patches between 2013 to 2019

| Year | Evaluation Index | | |
|---|---|---|---|
| | R | RMSE ($m^3/m^3$) | MAE ($m^3/m^3$) |
| 2013 | 0.974 | 0.065 | 0.044 |
| 2014 | 0.963 | 0.073 | 0.052 |
| 2015 | 0.968 | 0.069 | 0.050 |
| 2016 | 0.972 | 0.067 | 0.046 |
| 2017 | 0.966 | 0.070 | 0.049 |
| 2018 | 0.970 | 0.065 | 0.046 |
| 2019 | 0.969 | 0.069 | 0.048 |
| Average | 0.968 | 0.068 | 0.471 |



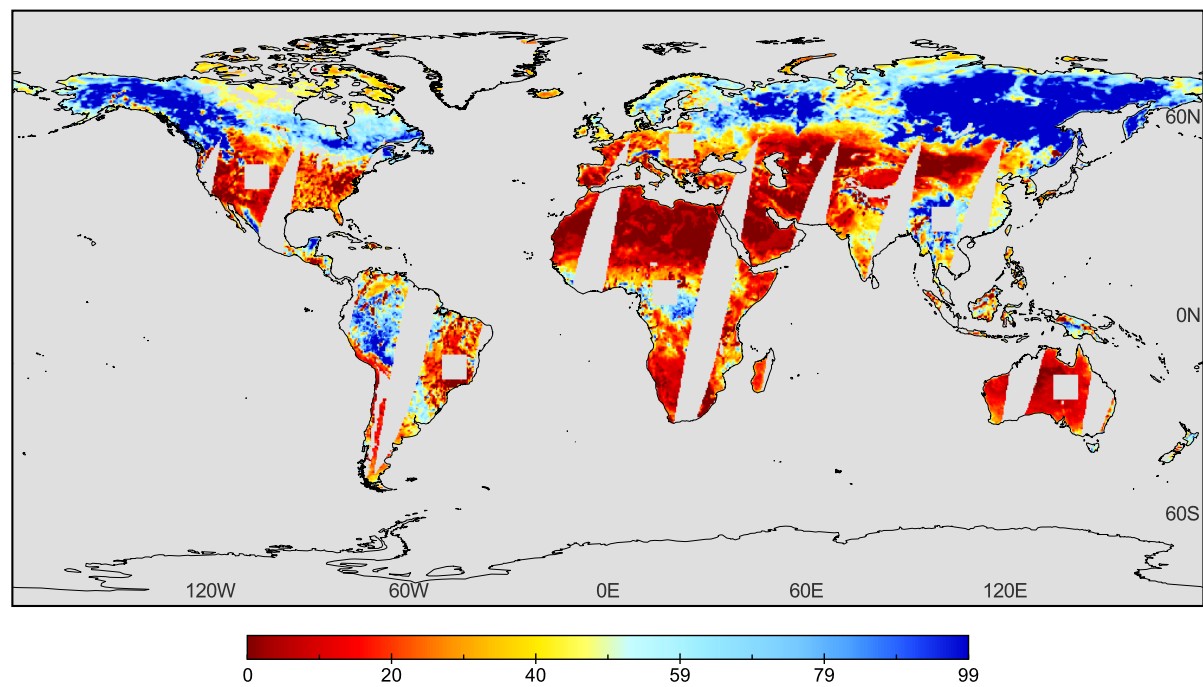

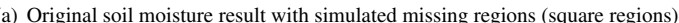

(a) Original soil moisture result with simulated missing regions (square regions)

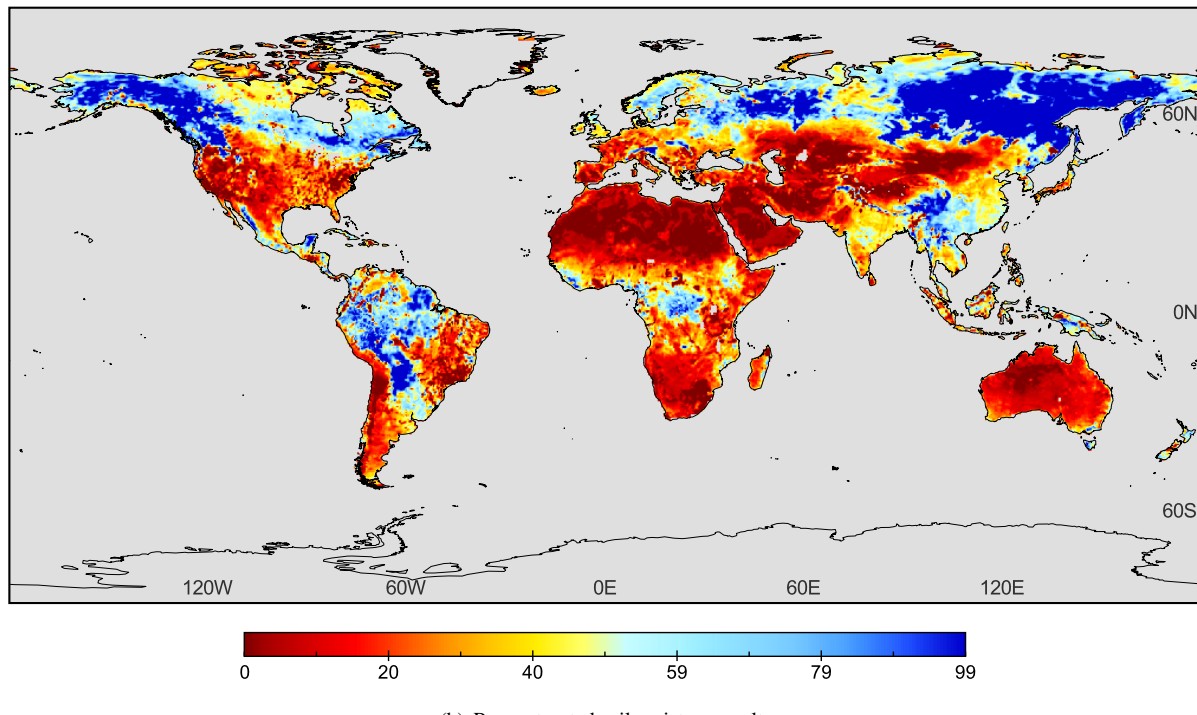

(b) Reconstructed soil moisture result

**Figure 9.** Original and reconstructed results with simulated missing regions in 2019.7.25



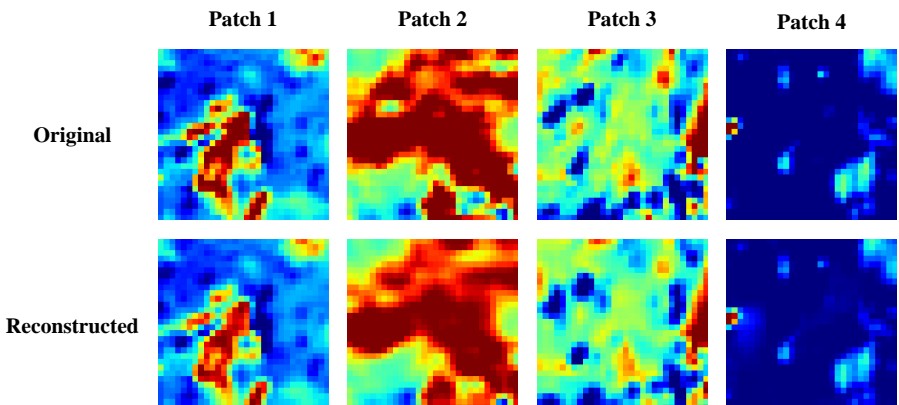

**Figure 10.** Detailed original/reconstructed spatial information of four simulated patches in 2015.7.25

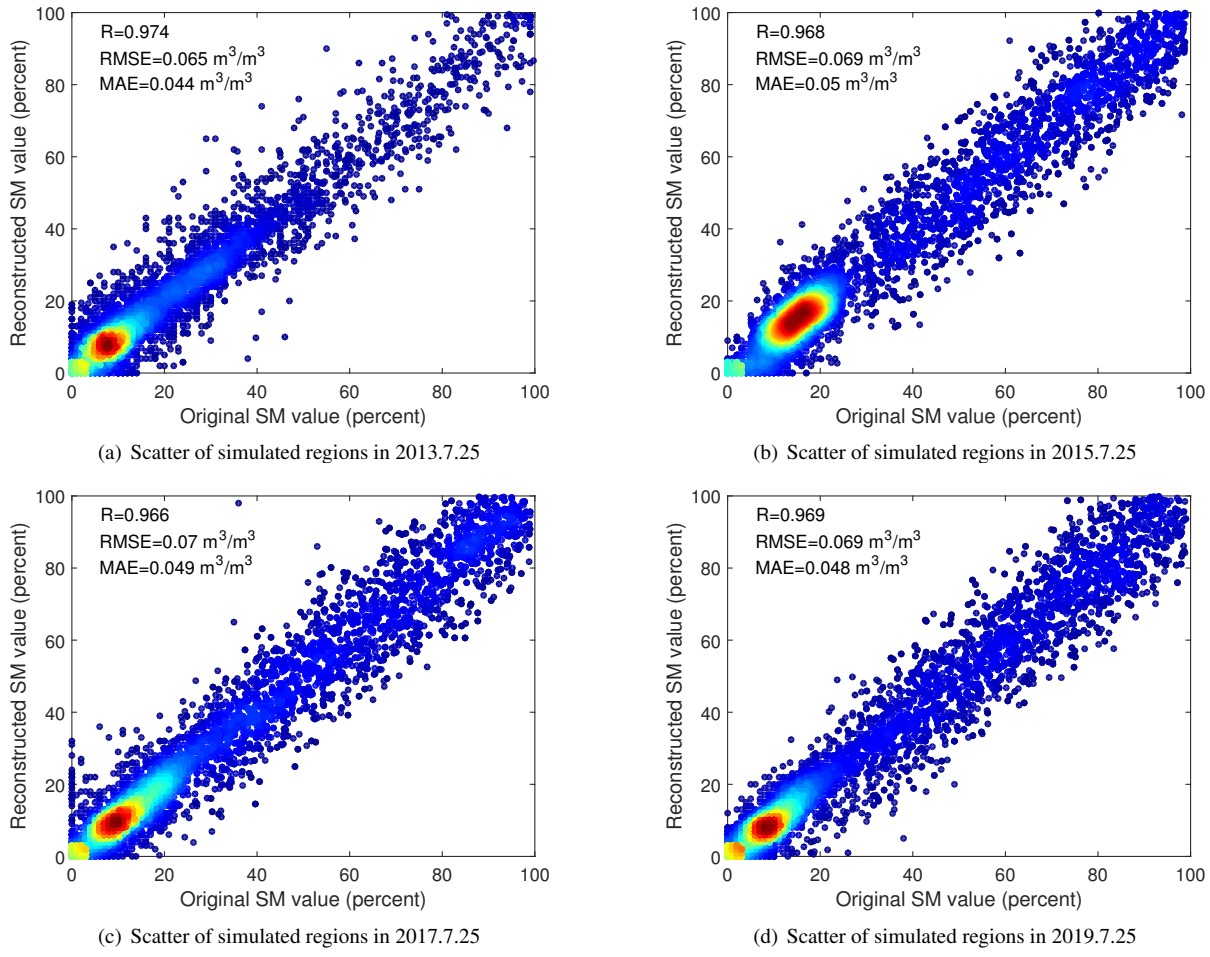

**Figure 11.** Original-reconstructed scatter of simulated regions in 2013, 2015, 2017, and 2019.7.25





## 5 Discussion

As mentioned in Section 1, some simple strategies such as time-series averaging can be also employed for synthesizing the complete soil moisture productions. Therefore, we perform the comparisons between the time-series averaging approach and the presented method, to further validate the effectiveness and rationality of our dataset and framework. In terms of the time-series averaging method, it averages the time-series daily soil moisture data to reconstruct gap regions. The original soil moisture result, time-series averaging result, and the proposed reconstructing result in 2016.9.10 are shown in Fig. 12(a)-(c), respectively. Three reconstructed regions are marked with black circle in Fig. 12(b) and (c). Besides, the evaluation index comparisons between the time-series averaging method and proposed method are listed in Table 3 through the corresponding in-situ data validations.

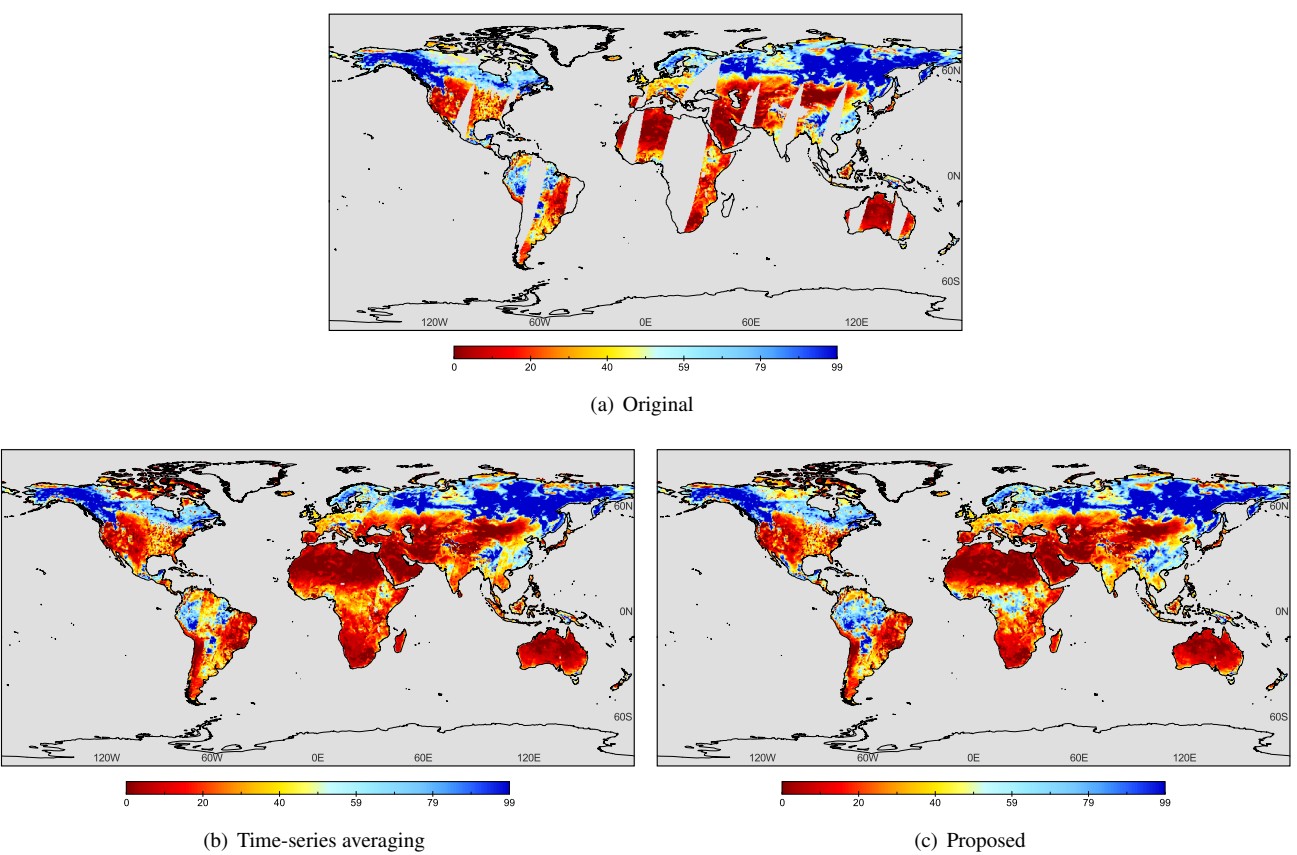

**Figure 12.** Original/time-series averaging/proposed global soil moisture results in 2016.9.10

As displayed in the balck circled regions of Fig. 12(b) and (c), we can clearly distinguish the spatial discontinuity in the time-series averaging result. Reversely, the proposed method performs better on spatial continuity between the valid and invalid regions. The evaluation indexes R, RMSE, and MAE also manifest the superiority of the presented approach, compared with the time-series averaging method in Table 3. The main reason is that daily soil moisture productions exist temporal dif-





**Table 3.** Evaluation index (R, RMSE, MAE) comparisons between the time-series averaging and proposed method

| Method | Evaluation Index | | |
|:---:|:---:|:---:|:---:|
| | R | RMSE ($m^3/m^3$) | MAE ($m^3/m^3$) |
| Time-series averaging | 0.635 | 0.124 | 0.093 |
| Proposed | 0.708 | 0.085 | 0.066 |

ferences. While the time-series averaging strategy cannot use the 2D-spatial context information and ignores these temporal differences. Therefore, it reflects the obvious "boundary difference effect" especially in the circled regions of Fig. 12(b). This also reveals the limitations and shortages of the time-series averaging method. On the contrary, the proposed method jointly utilizes both spatial and temporal information of these time-series soil moisture productions. Further, it can better richly exploit the deep spatio-temporal feature for soil moisture data reconstructing. Overall, this discussion demonstrates the superiority of

the proposed framework on time-series productions daily reconstructing, especially compared with the time-series averaging strategy.

## 6 Conclusions

In this work, aiming at the spatial incompleteness and temporal incontinuity, we generate a seamless global daily (SGD) AMSR2 soil moisture long-term productions from 2013 to 2019. To jointly utilize spatial and temporal information, a novel

3-D spatio-temporal partial CNN is proposed for AMSR2 soil moisture productions gap-filling. The partial 3D-CNN and global-local loss function are developed for better extracting valid region features and ignoring invalid regions through data and mask information. Three validation strategies are employed to testify the precision of our seamless global daily productions as follows: 1) In-situ validation; 2) Time-series validation; And 3) simulated missing regions validation. Evaluating results demonstrate that the seamless global daily AMSR2 soil moisture dataset shows high accuracy, reliability, and robustness.

Although the proposed framework performs well on generating this seamless global daily soil moisture dataset, some drawbacks and limitations still need to be overcome especially on multi-source data fusion, spatio-temporal information extracting and deep learning model optimization. In our future work, we will introduce the multi-source heterogeneous data such as daily rainfall and land cover information into the SGD framework. The proposed reconstructing model will be increasingly improved by means of more powerful units and structures. In addition, other global surface productions can also be considered such as

vegetation index and land surface temperature through the proposed reconstructing model.

*Data availability.* This dataset can be downloaded at **https://zenodo.org/record/3960425** (Zhang et al., 2020. DOI: 10.5281/zenodo.3960425). Besides, more details and experimental results of this work are displayed at https://qzhang95.github.io/Projects/Global-Daily-Seamless-AMSR2/ because of the paper layout limitations.



*Code availability.* The related codes (Python) of this work are also available at https://github.com/qzhang95/SGD-SM.

*Author contributions.* ZQ designed the proposed model and performed the experiments. YQQ and ZLP revised the whole manuscript. LJ, WY, and SFJ provided related data and some figures of this work. All authors read and provided suggestions for this manuscript.

*Competing interests.* All authors proclaim that they have no conflict of interests.

*Acknowledgements.* We sincerely thank ISMN organization, for them supplying scientific sites data.



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
