# Peer review of "Generating Seamless Global Daily AMSR2 Soil Moisture (SGD-SM) Long-term Products 2013-2019"

_Earth System Science Data, 2020_

## Referee Comment (RC1) · Anonymous Referee #1 · 1 Dec 2020

In this manuscript, authors generated a seamless global daily soil moisture dataset from 2013 to 2019. The incomplete daily global AMSR-2 soil moisture productions indeed exist the coverage problem, due to the satellite orbit coverage and the limitations of soil moisture retrieving algorithms. Overall, the proposed 3D spatio-temporal deep learning model is novelty for reconstructing the invalid soil moisture area, to solve the above coverage issue in AMSR-2 global daily products. In addition, three validation programs are employed in this manuscript to ensure the reliability of the seamless global daily soil moisture dataset. Several suggestions may be helpful to better improve this meaningful work:

[Figure]

1) How to deal with the unique mutations for the proposed reconstructing model, such as precipitation or snowfall in single day? It seems that this work relies on the sequential time-series redundancy for generating seamless global daily AMSR-2 soil moisture productions.

2) In the testing stage of Fig. 3, the convergence of the training spatio-temporal 3-D reconstructing model is vital for subsequent processing. Therefore, descriptions of this convergence condition must be illustrated in this manuscript.

3) Why authors use 3D partial convolutional neural network, rather than common convolutional neural network, for the soil moisture gap-filling task in missing area? Besides, the intentions for mask updating operation in the reconstructing model should be given.

4) Authors employed both local and global soil moisture information to optimize the network. The distinction and connection between local and global information need to provide the explanations and effects.

5) In the time-series validation, most of the soil moisture time-series scatters can obviously reveal the annual periodic variations in Fig. 8. Authors should take advantage of these annual periodic variations to better verify the rationality of the daily SM products.

6) In the simulated missing regions validation, the spatial continuity is also important for the reconstructed seamless soil moisture productions. To better distinguish the spatial details of reconstructed soil moisture, authors selected some enlarged patches in Fig. 10. More descriptions should be introduced to investigate this key point for spatial consistency between the reconstructing and adjacent regions.

---

## Referee Comment (RC2) · Anonymous Referee #2 · 11 Dec 2020

Soil moisture information from remote sensing is of great value to understand the interactions between the land and the atmosphere, drought evaluation, ecosystems, hydrology, and water resources. Data gaps of remotely sensed surface soil moisture due to orbits and other sensor and environmental factors in space and time hinder our understanding of these important phenomena, studies, and applications. To address this important issue, the authors have proposed an approach that wisely utilizes 3D spatio-temporal partial convolutional neural network (CNN), to exact both spatial and temporal information for global daily soil moisture product gap-filling. Moreover, the experimental results and related validation have been fully examined and implemented, making the results and quality of the generated data sets convincing. Overall, this work is interest-

ing and significant for generating seamless global daily (SGD) soil moisture products that could be valuable in a broad range of research and applications. I recommend acceptance of this manuscript into the prestigious journal of ESSD after addressing issues as follows:

Major issues: -The authors also need to emphasize the unique aspects of daily soil moisture products, compared with monthly/annual average soil moisture results at lower temporal resolution. The high temporal resolution and the global scale are the most important attributes and contributions of the generated SGD long-term soil moisture in this work. This is valuable in hydrology and climate communities.

-In Fig. 3, how did the authors design the patch selecting and mask simulating operations in the training procedure? In addition, in the testing procedure, it seems that the proposed model just uses 8-day soil moisture products. Why not use 16-day or 30-day products for gap-filling?

-In the validation section, the authors employed three validation approaches to test out the effectiveness of the SGD soil moisture production between 2013 to 2019: 1) In-situ validation; 2) Time-series validation; 3) Simulated missing regions validation. More explanations may need to be supplemented for these validations from both the spatial and temporal prospects.

-In the discussion section, the authors claimed that time-series averaging strategy has the obvious "boundary difference effect". And the contrast experiments are performed in Fig. 12 (b) and (c). What is the fundamental reason if better describing this common phenomenon, especially for monthly/annual average soil moisture products?

Minor issues: -Line 70: "for AMSR2 soil moisture productions gap-filling" may be better presented as "for global daily AMSR2 soil moisture productions gap-filling".

-Line 114: "part" may be better written as "a portion of".

-Line 232: "ignore the regions of" may be written as "ignore the coverage of".

-Line 243: "soil moisture stations (0 10cm)" lacks "-" in this sentence.

-Line 266: "daily time-series date between 2013 to 2019" may be written as "daily time-series date between Jan 1 2013‒Dec 31 2019."

-Line 305: In Table 3, the best statistical metrics such as R, RMSE, and MAE could be highlighted, to better demonstrate the superiority compared with the time-series averaging method.

---

## Referee Comment (RC3) · Anonymous Referee #3 · 11 Dec 2020

The complete satellite-based soil moisture products in space and in long time series can be assimilated to land surface models to generate spatiotemporal soil moisture at the global scale for climate/weather predictions and surface physical property retrieval. In this paper, the author generated the seamless Global Daily Advanced Microwave Scanning Radiometer 2 (AMSR 2) Soil Moisture (SGD-SM) products by using the developed 3D spatiotemporal partial convolutional neural network (CNN), which filled the gap of AMSR2 soil moisture products due to limitations of satellite orbit coverage and soil moisture retrieval algorithms. Assessing the quality of SGD-SM products was carried out by means of in-situ validation, time-series validation and the validation in selected missing regions. Furthermore, it showed that the SGD-SM) products had

improved R and RMSE by comparisons to those based on the time-series averaging.

Although it is enough to understand what 'went on', the scientific and English expressions are poor. Authors need to first go through the whole manuscript and make it readable. Meanwhile, the literature review is not very related to the deep learning method that the authors mentioned and used in this paper. The methodology part is not clear enough to follow. Considering the important applications of the complete products at the global scale, this review suggests to reconsider the paper after major revisions.

Major and minor comments are listed in blow and others please find them in the attachment.

Major comments: 1. Please revise the title. See the attachment.

2. Please give the definition of 'context information' and 'context consistency' used in this paper.

3. In lines 44-45, please explain who is 'the best observed value'. Please confirm 'a best single-point' or 'best single-points'.

4. In line 51, please give the definition of 'invalid land regions'.

5. In lines 55-58, please briefly introduce the advantage/weakness of the mentioned methods in the reference for fillings gaps of soil moisture products. The current literature review is just like a list and not informative to induce the developed method that you used in your study.

6. In lines 64-69, information like 'a new strategy to solve incomplete...obtain the global gap-filling' express the same meaning. The content in a), b) and c) sounds casual and is not concise in the scientific meaning. Most importantly, please state the reason why do you use the current deep learning method, although we know it is a hot topic. Since you mentioned deep learning theory, can authors give a literature review of soil moisture product gap-filling? I suggest to rewrite lines 48-65 to present a better

literature review and the motivation of your work.

7. In line 70, please explain why the AMSR2 soil moisture products are focused, such as its availability in long time series compared to other satellite soil moisture products.

8. In lines 70-83, it seems that 'a novel 3-D spatiotemporal partial convolutional neural network, global-local loss function' appears suddenly. I suggest to briefly explain them a bit when they are first mentioned. Meanwhile, the objective part presents the content in the Conclusions. They are different, please revise.

9. In line 97, please specify the uncertainty of soil moisture. What do you really refer to? Is it the uncertainty from the soil moisture retrieval algorithm or others?

10. In line 114, please give the spatial distribution of (the used) in-situ soil moisture networks.

11. In line 117, please do you mean descending and ascending data for 'neighboring in-situ hourly values'?

12. The Methodology part is not clear and neat. In line 125, what is 'the loss convergent model'. It appears also suddenly. I suggest to rewrite the overall descriptions of the method, and clearly explain every step and their relations in a logical way. Please present the following sections in a more clear way. There are lots of numbers mentioned, like T-4, T+4, 3*3*3 (what does 3 mean?), 11 layers, 90, 0.1 during the training procedure, 128, 210, 300, 0.001, etc. I question their rationalities, please give the reason for each. In line 190, I am not sure about the relation between loss function and learning parameters? By the way, who is the learning parameter in this study? In line 208, "After building up this unified loss function, the presented reconstructing model employs Adam algorithm as the gradient descent strategy, The number of batch size in this model is fixed as 128 for network training. The total epochs and initial learning 210 rate are determined as 300 and 0.001, respectively. Starting every 30 epochs, the learning rate is degraded through decay coefficient 0.5." Please explain a bit in a clear

way, it is very difficult for laymen to understand 'epoches, Adam algorithms and the gradient descent strategy'.

13. At the beginning of section 4, please put the doi related content in the section of 'Data availability'. Additionally, please remove the duplicate information that is already mentioned in the Method. Please only present your results in the Result section.

14. Figure 10, the original patch shows almost the same as the reconstructed. Do you mean the original patch is missing here? I am sorry if I misunderstand.

15. Figure 12, no black circles.

16. Please describe uncertainties in this generated SGD-SM product.

Minor comments 1. Please follow "Manuscript composition (https://www.earth-system-science-data.net/submission.html#manuscriptcomposition)" to make all related, e.g., Data availability as a separate section and use Sect accord with regulations.

2. Use 'besides' too many times in a scientific paper.

3. In line 20, I do not think EAS CCI is a sensor. Please revise.

4. 'products' not 'Production'

Please also note the supplement to this comment:
https://essd.copernicus.org/preprints/essd-2020-353/essd-2020-353-RC3-supplement.pdf

**Supplement:**

[revised manuscript text omitted]

---

## Referee Comment (RC4) · Anonymous Referee #4 · 31 Dec 2020

The authors present a method and dataset to fill daily AMSR2 soil moisture product gaps with a CNN for the years 2013-2019.

The abstract jumps quickly into the topic, but is somehow ambiguous by not being clear to which current soil moisture products (". . . the acquired daily soil moisture productions") the introduction relates? It would be good to explicitly state that it relates to the AMSR2 products (not productions) . . . this lack of language clarity (e.g. "reliable cooperativity. . ." etc.) traces through the whole manuscript and needs to be strictly revised before considering acceptance. Otherwise it is really hard for the reader to understand, and thus, to estimate the usefulness of this dataset. However, independent

of the language a few other comments as follows:

Still the abstract creates more questions than answers. The evaluation measures are difficult to interpret. Why stating 2 evaluation measures, with one for original data? Also the choice of units (m3/m3) is not immediately clear as the dataset only produces percent values for soil moisture?

The introduction is not clearly introducing the AMSR2 original dataset. I think it would be of great value if at least basic technical cornerstone of the original dataset are described.

Stating that the codes are also published is misleading so far. "The related Python codes of this dataset are also available at https://github.com/qzhang95/SGD-SM." (authors) only holds an example of code of extracting data. I think it would be really helping the transparency of the data quality if the physical network implementation (TensorFlow, Keras, SKLearn, PyTorch?) would be also open in the spirit of open data and open source and reproducibility. Understandably, a trained neural network model is not 100% reproducible, but the model could be also archived on Zenodo? It only makes sense, because it would be very feasibile to update the dataset on a yearly basis with the developed model. In the end the idea of ESSD is "living data".

The year 2013 folder only contains 362 files, not 365. May 2013 only seems to have 28 files? Please check your upload on Zenodo.

(Zhang et al., 2020) This citation is not in the references list, there are only Zhang 2020a and 2020b. Please add, presumably it is your data citation: DOI: 10.5281/zenodo.396042

"More details of this work are released at https://qzhang95.github.io/Projects/Global-Daily-Seamless-AMSR2/." ... your current paper should reflect the most important and up to date source of information until it is published. It would be ok though to refer to it as a "technical supplement" maybe? However, those URLs are not reliable. Thus, if

you have a technical supplement with more details, it could be added to your Zenodo archive (which will be reliable and has a DOI).

Section 3.1 starting on page 7 is the methodological main part of your neural network implementation. While I'd like to acknowledge that technical level of description, I have two contradicting issues with it:

1) p7 ll137-139: "This network includes 11 layers (3D partial CNN unit and ReLU (Rectified Linear Unit)) in Fig. 4. The size of 3D filters isall set as $3 \times 3 \times 3$. Number of feature maps before ten layers is fixed as 90, and the channel of feature map in the final layer is exported as 1".

That is very technical, yet, it is not clear why those dimensions where chosen. The discussion section does not discuss the CNN and the design choices at all and what effect they have. For example, how can this capture the comparatively big gap areas of the original AMSR2 dataset?

2) On the other hand, much of this could also go into a technical supplement and you could provide a much higher-level overview for the reader in the paper. The paper is the data description, and many readers and future users of the dataset will not have the technical understanding of judging or even reading through the technical low-level design of the CNN – nevertheless this still also needs to be documented.

Also, why not an LSTM type network?

Calling it spatio-temporal 3D might be misleading, as it is areal 2D and then a temporal dimension. Spatio-temporal indicates that already, the added 3D might lead to think of spatial 3D plus time.

What was the reason to chose the 0.25 dec degrees as spacing for the data files?

Last but not least, I'd like to advocate for a bit more metadata in the netcdf files, because netcdf provides great means for metadata. For example, you could adhere a bit more to the NetCDF-CF conventions, or at least add e.g. attributes such as title,

reference and a time stamp in the dataset, not relying on the filename for example. You could also join at least the yearly slices into a "cube" that follows conventions of the Earth Sciences community (e.g. longitude instead of lon as variable name). Also the Zenodo deposit could have more fields filled out for improved discovery, more keywords (e.g. "soil moisture"?) and terms from controlled vocabularies, such as GEMET (https://www.eea.europa.eu/help/glossary/gemet-environmental-thesaurus ) or similar.

I think there is a lot of potential in this paper and dataset (even if "only" 6 years length). However, many technical aspects make it hard to grasp the content and estimate the quality in the first place.

---

## Author Comment (AC1) · 12 Jan 2021

**Response to the Comments of Referee #1**

Dear Referee #1:

We are particularly grateful for your careful reading, and for giving us many constructive comments of this work!

According to the comments and suggestions, we have tried our best to improve the previous manuscript essd-2020-353 (SGD-SM: Generating Seamless Global Daily AMSR2 Soil Moisture Long-term Productions (2013–2019)). An item-by-item response follows.

Once again, we are particularly grateful for your careful reading and constructive comments. Thanks very much for your time.

Best regards,

Qiang Zhang

**General comments:**

*In this manuscript, authors generated a seamless global daily soil moisture dataset from 2013 to 2019. The incomplete daily global AMSR-2 soil moisture productions indeed exist the coverage problem, due to the satellite orbit coverage and the limitations of soil moisture retrieving algorithms. Overall, the proposed 3D spatio-temporal deep learning model is novelty for reconstructing the invalid soil moisture area, to solve the above coverage issue in AMSR-2 global daily products. In addition, three validation programs are employed in this manuscript to ensure the reliability of the seamless global daily soil moisture dataset. Several suggestions may be helpful to better improve this meaningful work.*

**Response:** We are particularly grateful to the reviewer for his/her detailed suggestions! According to the comments, we have tried our best to improve the previous manuscript. An item-by-item response to each constructive comment follows.

**Major comments:**

**Q1.1:** *How to deal with the unique mutations for the proposed reconstructing model, such as precipitation or snowfall in single day? It seems that this work relies on the sequential time-series redundancy for generating seamless global daily AMSR-2 soil moisture products.*

**Response:** Thanks for this issue. As the referee stated, this work relies on the sequential time-series redundancy for generating SGD-SM products. For unique mutations, it also influences the latter days due to the inertial effects. Therefore, we take the $T$-4 to $T$+4 date SM as the input data, to utilize the continuity property of time-series values. In our future work, we will introduce multi-source information fusion into the proposed model, such as precipitation and snowfall.

**Q1.2:** *In the testing stage of Fig. 3, the convergence of the training spatio-temporal 3-D reconstructing model is vital for subsequent processing. Therefore, descriptions of this convergence condition must be illustrated in this manuscript.*

**Response:** Thanks for this meaningful suggestion. The convergence condition denotes that the loss of the proposed model gradually decreases, and finally maintains smooth in training procedure. We have supplemented this description in the revised manuscript.

**Q1.3:** *Why authors use 3D partial convolutional neural network, rather than common convolutional neural network, for the soil moisture gap-filling task in missing area? Besides, the intentions for mask updating operation in the reconstructing model should be given.*

**Response:** Thanks for this issue. it should be highlighted that the valid and invalid SM simultaneously exist especially around the coast regions and gap regions. The common CNN ignores the location information of invalid or valid pixels in soil moisture data, which cannot eliminate the invalid information. Therefore, to solve this negative effect, we develop the partial 3D-CNN to ignore the invalid information in the proposed reconstructing model. the partial convolutional output is only decided by the valid soil moisture pixels of input, rather than the invalid soil moisture pixels. Through the mask, we can effectively exclude the interference information of invalid soil moisture pixels such as marine regions and gap regions. Then the scaling divisor in Eq. (2) further adjusts for the variational number of valid soil moisture pixels. If the partial convolution can generate at least one valid value of the output result, then we mark this location as valid value in the new masks. We have added these explanations into the revised version.

**Q1.4:** *Authors employed both local and global soil moisture information to optimize the network. The distinction and connection between local and global information need to provide the explanations and effects.*

**Response:** Thanks for this comment. Euclidean loss function only pays attention to the holistic information bias for network optimization. It ignores the soil moisture particularity of the local areas, especially in local coastal, mountain, and hinterland regions. However, this particularity is extremely significant for invalid regions gap-filling, because of the spatial heterogeneity in soil moisture products. Therefore, to take both the global consistency and local soil moisture particularity into consideration, the global land mask and current mask in date $T$ are both employed after the final layer. Through this way, we can simultaneously ensure the global consistency and distinguish the local particularity. We have supplemented these descriptions in the revised manuscript.

**Q1.5:** *In the time-series validation, most of the soil moisture time-series scatters can obviously reveal the annual periodic variations in Fig. 8. Authors should take advantage of these annual periodic variations to better verify the rationality of the daily SM products.*

**Response:** Thanks for mentioning this issue. As depicted in Fig. 8(a)-(f), most of the soil moisture time-series scatters can obviously reveal the annual periodic variations. The reconstructed soil moisture results generally behave fine temporal consistency with the original soil moisture results in different areas. Related low soil moisture values mostly existed in the droughty season of winter with the frozen lands such as in Fig. 8(d). Related high soil moisture values mainly generated in the moist season of summer with more rainy days, especially in Fig. 8 (b), (d) and (f).

Overall, compared with the whole original variation tendency between 2013 to 2019, the generated seamless global daily AMSR2 soil moisture long-term products can steadily reflect the

temporal consistency and variation. It is significant for time-series applications and analysis. This daily time-series validation also demonstrates the robustness of the presented method and the availability of the established seamless global daily products. We have emphasized these annual periodic variations in the revised manuscript, for better verifying the rationality of the daily products.

**Q1.6:** *In the simulated missing regions validation, the spatial continuity is also important for the reconstructed seamless soil moisture productions. To better distinguish the spatial details of reconstructed soil moisture, authors selected some enlarged patches in Fig. 10. More descriptions should be introduced to investigate this key point for spatial consistency between the reconstructing and adjacent regions.*

**Response:** Many thanks for this suggestion! To better distinguish the spatial details of reconstructed soil moisture, we select four enlarged patches in simulated regions in Fig. 10. It can be clearly observed that the reconstructed patches perform the high consistency with the original patches, as displayed in Fig. 10. The reconstructed invalid regions are consecutive between the original valid regions. And in the simulated missing patches, the spatial texture information is also continuous without obvious boundary reconstructing effects. These descriptions have been introduced into the revised version.

[Figure]

**Fig. 10.** Detailed original/reconstructed spatial information of four simulated patches in 2015.7.25

---

## Author Comment (AC2) · 12 Jan 2021

**Response to the Comments of Referee #2**

Dear Referee #2:

We are particularly grateful for your careful reading, and for giving us many constructive comments of this work!

According to the comments and suggestions, we have tried our best to improve the previous manuscript essd-2020-353 (SGD-SM: Generating Seamless Global Daily AMSR2 Soil Moisture Long-term Productions (2013–2019)). An item-by-item response follows.

Once again, we are particularly grateful for your careful reading and constructive comments. Thanks very much for your time.

Best regards,

Qiang Zhang

**General comments:**

*Soil moisture information from remote sensing is of great value to understand the inter-actions between the land and the atmosphere, drought evaluation, ecosystems, hydrology, and water resources. Data gaps of remotely sensed surface soil moisture due to orbits and other sensor and environmental factors in space and time hinder our understanding of these important phenomena, studies, and applications. To address this important issue, the authors have proposed an approach that wisely utilizes 3D spatiotemporal partial convolutional neural network, to exact both spatial and temporal information for global daily soil moisture product gap-filling. Moreover, the experimental results and related validation have been fully examined and implemented, making the results and quality of the generated data sets convincing. Overall, this work is interesting and significant for generating seamless global daily (SGD) soil moisture products that could be valuable in a broad range of research and applications. I recommend acceptance of this manuscript into the prestigious journal of ESSD after addressing issues as follows.*

**Response:** We are particularly grateful to the reviewer for his/her approval and detailed suggestions! According to the comments, we have tried our best to improve the previous manuscript. An item-by-item response to each constructive comment follows.

**Major comments:**

**Q2.1:** *The authors also need to emphasize the unique aspects of daily soil moisture products, compared with monthly/annual average soil moisture results at lower temporal resolution. The high temporal resolution and the global scale are the most important attributes and contributions of the generated SGD long-term soil moisture in this work. This is valuable in hydrology and climate communities.*

**Response:** Thanks for this significant suggestion. For most applications and spatial analysis, the incomplete soil moisture products are overall averaged as the monthly/quarterly/yearly results to generate the complete products. This operation can effectively improve the spatial soil moisture coverage. However, it distinctly sacrifices the high-frequency temporal resolution as low-frequency temporal resolution, which also severely reduces the data utilization. Besides, it ignores the unique spatial distribution of single day and loses the dense time-series changing information.

From these perspectives, a novel 3-D partial convolutional neural network is proposed for AMSR2 soil moisture products gap-filling. By means of the proposed method, we can effectively break through the above-mentioned limitations. And finally, this work generates the seamless global daily AMSR2 soil moisture long-term products from 2013 to 2019.

> **Q2.2:** *In Fig. 3, how did the authors design the patch selecting and mask simulating operations in the training procedure? In addition, in the testing procedure, it seems that the proposed model just uses 8-day soil moisture products. Why not use 16-day or 30-day products for gap-filling?*

**Response:** Thanks for these comments. Detailed descriptions are listed below:

1) In the patch selecting step, we traverse the global regions in date $T$ to select the complete soil moisture patch label, whose local land regions are undamaged. It should be noted the rest incomplete patches in date $T$ are excluded because they cannot participate in the supervised learning. The corresponding time-series soil moisture patches of this selected patch between date $T$-4 to $T$+4, is set as the spatio-temporal data patch groups. And their corresponding masks between date $T$-4 to $T$+4 is set as the spatio-temporal mask patch groups. After traversing the original products from 2013 to 2019, we finally establish the spatio-temporal data and mask patch groups with the number of 276488 patches. The soil moisture patch size is fixed as 40×40 for patch selecting.

2) In the mask simulating step, 10000 patch masks of the size 40×40 are chosen from the

global AMSR2 soil moisture masks from 2013 to 2019. The missing ratio range of these masks is set as [0.3, 0.7]. Then these patch masks are randomly selected for label patches use within the spatio-temporal data and mask patch groups. The complete patch in date $T$ (label) is simulated as the incomplete patch (data) through the above mask. And the original corresponding mask of this patch needs also to be replaced. After traversing and building the label-data spatio-temporal patch groups, this dataset is set as the training samples for the usage of reconstructing network.

3) In terms of using 8-day soil moisture products not 16-day or 30-day for gap-filling, we mainly consider the adjacent rule. Generally, 8-day products have the most highly correlated relation, compared with 16-day or 30-day products. Therefore, we choose the 8-day products from the reliability and accuracy prospects.

**Q2.3:** *In the validation section, the authors employed three validation approaches to test out the effectiveness of the SGD soil moisture production between 2013 to 2019: 1) In-situ validation; 2) Time-series validation; 3) Simulated missing regions validation. More explanations may need to be supplemented for these validations from both the spatial and temporal prospects.*

**Response:** Thanks for this issue. In-situ validation is utilized to compare the reconstructed soil moisture with original AMSR2 soil moisture through the selected in situ sites from the spatial prospect. In-situ shallow-depth soil moisture sites can be employed as the ground-truth to validate the reconstructing satellite soil moisture products. Time-series validation is employed for evaluating the time-series continuity from the temporal prospect. Soil moisture time-series scatters can obviously reveal the annual periodic variations for time-series validation. Simulated missing regions validation is used to testify the soil moisture consistency from the spatial prospect. It can verify the spatial consistency between the valid and invalid soil moisture regions. We have added these explanations into the revised manuscript.

**Q2.4:** *In the discussion section, the authors claimed that time-series averaging strategy has the obvious "boundary difference effect". And the contrast experiments are performed in Fig. 12 (b) and (c). What is the fundamental reason if better describing this common phenomenon, especially for monthly/annual average soil moisture products?*

**Response:** Thanks for this meaningful query. The time-series averaging strategy ignores the unique spatial distribution of single day and loses the dense time-series changing information. In other word, the monthly/quarterly/yearly soil moisture data averaging operations damage the initial information on both spatial and temporal dimension. The time-series averaging strategy cannot use the 2D-spatial information and neglects these temporal differences. Therefore, it reflects the obvious "boundary difference effect", as shown in Fig. 12(a). This also reveals the limitations and shortages of the time-series averaging method. On the contrary, the proposed method jointly utilizes both spatial and temporal information of these time-series soil moisture products. Further, the proposed method can better richly exploit the deep spatio-temporal feature for soil moisture data reconstructing, as shown in Fig. 12(b). We have supplemented these reasons in the revised version.

[Figure]

(a) Time-series averaging            (b) Proposed

**Figure 12.** Original/time-series averaging/proposed global soil moisture results in 2016.9.10

**Minor comments:**

**Q2.5:** *Line 70: "for AMSR2 soil moisture productions gap-filling" may be better presented as "for global daily AMSR2 soil moisture productions gap-filling".*

**Response:** We have revised this sentence as the referee's suggestion.

**Q2.6:** *Line 114: "part" may be better written as "a portion of".*

**Response:** We have revised this sentence as the referee's suggestion.

**Q2.7:** *Line 232: "ignore the regions of" may be written as "ignore the coverage of".*

**Response:** We have revised this sentence as the referee's suggestion.

**Q2.8:** *Line 243: "soil moisture stations (0 10cm)" lacks "-" in this sentence.*

**Response:** We have added "-" into this sentence.

**Q2.9:** *Line 266: "daily time-series date between 2013 to 2019" may be written as "daily time-series date between Jan 1 2013 to Dec 31 2019."*

**Response:** We have revised this sentence as the referee's suggestion.

**Q2.10:** *Line 305: In Table 3, the best statistical metrics such as R, RMSE, and MAE could be highlighted, to better demonstrate the superiority compared with the time-series averaging method.*

**Response:** We have highlighted the best statistical metrics in Table 3 as follow:

Table 3. Evaluation index comparisons between the time-series averaging and proposed method

| Method | Evaluation index | | |
|---|---|---|---|
| | R | RMSE | MAE |
| Time-series averaging | 0.635 | 0.124 | 0.093 |
| Proposed | **0.708** | **0.085** | **0.066** |

---

## Author Comment (AC3) · 12 Jan 2021

**Response to the Comments of Referee #3**

Dear Referee #3:

We are particularly grateful for your careful reading, and for giving us the constructive comments of this manuscript!

According to the comments and suggestions, we have tried our best to improve the previous manuscript essd-2020-353 (SGD-SM: Generating Seamless Global Daily AMSR2 Soil Moisture Long-term Productions (2013–2019)). An item-by-item response follows.

Once again, we are particularly grateful for your careful reading and constructive comments. Thanks very much for your time.

Best regards,

Qiang Zhang

**General comments:**

*The complete satellite-based soil moisture products in space and in long time series can be assimilated to land surface models to generate spatiotemporal soil moisture at the global scale for climate/weather predictions and surface physical property retrieval. In this paper, the author generated the seamless Global Daily Advanced Microwave Scanning Radiometer 2 (AMSR2) Soil Moisture (SGD-SM) products by using the developed 3D spatiotemporal partial convolutional neural network (CNN), which filled the gap of AMSR2 soil moisture products due to limitations of satellite orbit coverage and soil moisture retrieval algorithms. Assessing the quality of SGD-SM products was carried out by means of in-situ validation, time-series validation and the validation in selected missing regions. Furthermore, it showed that the SGD-SM products had improved R and RMSE by comparisons to those based on the time-series averaging. Although it is enough to understand what 'went on', the scientific and English expressions are poor. Authors need to first go through the whole manuscript and make it readable. Meanwhile, the literature review is not very related to the deep learning method that the authors mentioned and used in this paper. The methodology part is not clear enough to follow. Considering the important applications of the complete products at the global scale, this review suggests to reconsider the paper after major revisions.*

*Major and minor comments are listed in blow and others please find them in the attachment.*

**Response:** We are particularly grateful to the referee for his/her careful reading and detailed suggestions! For the language clarity, we have revised the whole manuscript sentence by sentence in the updated version. The literature review of this work has been rewritten in Q3.6. According to the comments, we have tried our best to improve the previous manuscript. An item-by-item response to each constructive comment follows.

**Major comments:**

**Q3.1:** *Please revise the title. See the attachment.*

**Response:** Thanks for this significant suggestion. We have revised the title as: 'Generating Seamless Global Daily AMSR2 Soil Moisture (SGD-SM) Long-term Products 2013-2019'.

**Q3.2:** *Please give the definition of 'context information' and 'context consistency' used in this paper.*

**Response:** Thanks for this comment. For avoiding understanding in this work, we have revised these two expressions 'context information' and 'context consistency', as 'original information' and 'spatial consistency', respectively.

**Q3.3:** *In lines 44-45, please explain who is "the best observed value". Please confirm "a best single-point" or "best single-points".*

**Response:** Thanks for this issue. We have corrected these problematic descriptions in multi-temporal soil moisture data synthesizing. 'the best observed value' has been replaced with 'the valid value'. 'best single-point' has been revised as 'valid single-point'.

**Q3.4:** *In line 51, please give the definition of 'invalid land regions'.*

**Response:** Thanks for this suggestion. The 'invalid land regions' refers to the gap or information missing area. We have supplemented this definition in current manuscript.

**Q3.5:** *In lines 55-58, please briefly introduce the advantage/weakness of the mentioned methods in the reference for fillings gaps of soil moisture products. The current literature review is just like a list and not informative to induce the developed method that you used in your study.*

**Response:** Thanks for this beneficial comment. We have introduced the advantage/weakness of the mentioned methods in the reference for fillings gaps of soil moisture products as follow:

'Overall, these methods can effectively fill the gaps of soil moisture products. However, these methods cannot simultaneously take both spatial and temporal information into consideration. In addition, the daily soil moisture products in global scale have not been exploited up to now.'

**Q3.6:** *In lines 64-69, information like 'a new strategy to solve incomplete...obtain the global gap-filling' express the same meaning. The content in a), b) and c) sounds casual and is not concise in the scientific meaning. Most importantly, please state the reason why do you use the current deep learning method, although we know it is a hot topic. Since you mentioned deep learning, can authors give a literature review of soil moisture product gap-filling? I suggest to rewrite lines 48-65 to present a better literature review and the motivation of your work.*

**Response:** We are very grateful for these significant suggestions on literature review! To better demonstrate the motivation of this work, we have rewritten the literature review for oil moisture products gap-filling as follow:

'To overcome above-mentioned limitations, some missing values reconstruction methods have been developed especially on multi-temporal images thick cloud removal and deadline gap-filling (Zhang et al., 2020a). For example, Zhu et al. (2011) proposed the multi-temporal neighboring homologous value padding method for thick cloud removal. Chen et al. (2011) presented an effective interpolating algorithm for recovering the invalid regions in Landsat images. Zhang et

al. (2018a) built an integrative spatio-temporal-spectral network for missing data reconstruction in multiple tasks.

In terms of the soil moisture products gap-filling, several methods have also been proposed to address this issue. Wang et al. (2012) presented a penalized least square regression-based approach for global satellite soil moisture gap filling observation. Fang et al. (2017) introduced a long short-term memory network to generate spatial complete overlay SMAP in U.S. Long et al. (2019) fused multi-resolution soil moisture products, which can produce daily fine-resolution data in local regions. Llamas et al. (2020) used geostatistical techniques and multiple regression strategy to get spatial complete results of satellite-derived products. Overall, there are few works for soil moisture productions reconstructing on global and daily scale.

In spatial dimension, the invalid land areas and adjacent valid land areas exist the spatial consistency and spatial correlation on daily soil moisture products (Long et al., 2020). In temporal dimension, daily time-series changing curve of the same point natively appears with the continuous and smooth peculiarities (Chan et al., 2018). Overall, these methods can effectively fill the gaps of soil moisture products. However, these methods cannot simultaneously take both spatial and temporal information into consideration. In addition, the daily soil moisture products in global scale have not been exploited up to now.

Therefore, how about simultaneously extracting both spatial and temporal features for seamless global daily soil moisture products gap-filling? Recently, deep learning has gradually revealed the potential for remote sensing products processing (Chen et al., 2021). In consideration of the powerful feature expression ability via deep learning, can we utilize spatio-temporal information to generate long-term soil moisture products?'

**Q3.7:** *In line 70, please explain why the AMSR2 soil moisture products are focused, such as its availability in long time series compared to other satellite soil moisture products.*

**Response:** Thanks for this comment. The reason why the AMSR2 soil moisture products are focused in this work is discribed as follow:

'In consideration of the global coverage, temporal-resolution, and current availability, we select AMSR2 soil moisture products as the focused object.' In our future work, we will consider more soil moisture products such as AMSR-E, SMOS-IC, SMAP and so on.This explanation has been supplemented in the revised manuscript.

**Q3.8:** *In lines 70-83, it seems that 'a novel 3-D spatiotemporal partial convolutional neural network, global-local loss function' appears suddenly. I suggest to briefly explain them a bit when they are first mentioned. Meanwhile, the objective part presents the content in the Conclusions. They are different, please revise.*

**Response:** Thanks for this helpful suggestion! We have revised these sentences as 'a novel 3-D spatio-temporal deep learning framework is proposed for AMSR2 soil moisture products gap-filling.' and 'To optimize the proposed network, we develop a global-local loss function for excluding the invalid information.'

In addition, we have also rewritten the conclusions part to keep consistent with the objective part as follow:

'In this work, aiming at the spatial incompleteness and temporal incontinuity, we generate a seamless global daily (SGD) AMSR2 soil moisture long-term products from 2013 to 2019. To jointly utilize spatial and temporal information, a novel spatio-temporal partial CNN is proposed for AMSR2 soil moisture products gap-filling. The partial 3D-CNN and global-local loss function are developed for better extracting valid region features and ignoring invalid regions through data and mask information. Three validation strategies are employed to testify the precision of our seamless global daily products as follows: 1) In-situ validation; 2) Time-series validation; And 3) simulated

missing regions validation. Evaluating results demonstrate that the seamless global daily AMSR2 soil moisture dataset shows high accuracy, reliability, and robustness.'

**Q3.9:** *In line 97, please specify the uncertainty of soil moisture. What do you really refer to? Is it the uncertainty from the soil moisture retrieval algorithm or others?*

**Response:** Thanks for this query. The uncertainty of soil moisture refers to the LPRM-AMSR2 data variable "soil_moisture_c1_error". This uncertainty is generated by LPRM retrieval algorithm in daily soil moisture products. We have added this explanation into the updated manuscript.

**Q3.10:** *In line 114, please give the spatial distribution of (the used) in-situ soil moisture networks.*

**Response:** Thanks for this helpful comment. The spatial distribution of the used in-situ sites is depicted as below:

[Figure]

**Figure A.** The spatial distribution of the used in-situ sites.

**Q3.11:** *In line 117, please do you mean descending and ascending data for 'neighboring in-situ hourly values'?*

**Response:** Thanks for this query. 'neighboring in-situ hourly values' means that to validate the proposed SGD-SM products through in-situ validation, we must match the remote-sensing SM data with in-situ data nearly at the same time. Because in-situ values are the hourly data, we cannot obtain the coincident in-situ data for current date AMSR2 descending SM. Therefore, we select the two neighboring in-situ hourly SM values of AMSR2 SM (e.g., AMSR2 Descending data at 01:20, the neighboring in-situs are selected at 1:00 and 2:00). Then the two neighboring in-situ hourly values are averaged as the ultimate result of current date.

**Q3.12:** *The Methodology part is not clear and neat. In line 125, what is 'the loss convergent model'. It appears also suddenly. I suggest to rewrite the overall descriptions of the method, and clearly explain every step and their relations in a logical way. Please present the following sections in a clearer way. There are lots of numbers mentioned, like T-4, T+4, 3\*3\*3 (what does 3 mean?), 11 layers, 90, 0.1 during the training procedure, 128, 300, 0.001, etc. I question their rationalities, please give the reason for each. In line 190, I am not sure about the relation between loss function and learning parameters? By the way, who is the learning parameter in this study? In line 208, "After building up this unified loss function, the presented reconstructing model employs Adam algorithm as the gradient descent strategy, the number of batch size in this model is fixed as 128 for network training. The total epochs and initial learning rate are determined as 300 and 0.001, respectively. Starting every 30 epochs, the learning rate is degraded through decay coefficient 0.5." Please explain a bit in a clear way, it is very difficult for laymen to understand 'epochs, Adam algorithms and the gradient descent strategy'.*

**Response:** Many thanks for these meaningful suggestions! Deep learning allows computational models that are composed of multiple processing layers, to learn representations of data with multiple levels of abstraction. The forward-propagation and back-propagation are employed for optimizing the trainable parameters in neural network. I suggest referee can read the classical article (Yann LeCun et al., Deep Learning, *Nature*, 2015), to further understand more concepts in deep learning. Detailed explanations are listed as follows:

1) What is 'the loss convergent model': The loss convergence model denotes that the loss of the proposed model gradually decreases, and finally maintains smooth in training procedure. We have supplemented this description in the revised manuscript.

2) Overall descriptions of the method: We have rewritten the overall descriptions of the method, and clearly explain every step and their relations in a logical way.

3) Reason for each number: '$T$' stands for current daily date. '3×3×3' refers to the kernel size of 3D convolutional cube filter. '11 layers' represents the depth of the proposed deep neural network. '90' is the feature map number in CNN. '0.1' denotes the balancing factor to adjust the local loss and global loss in Eq. (6). '128' stands for the batch size in deep learning model. '0.001' refers to the learning rate for the training procedure.

4) Relation between loss function and learning parameters: In deep learning theory, the loss function is the 'baton' of the whole network, which guides the network parameters learning through the error back-propagation between the predicted sample and the original sample. In terms of the learning parameters, they represent the weighted and bias parameters in all the layers.

5) How to understand 'epochs, Adam algorithms and the gradient descent strategy': One 'Epoch' represents that the network goes through all the training data. 'Adam algorithms' is a gradient descent method in back-propagation step, to optimize the whole network parameters. 'gradient descent' denotes the partial differentiation and then updates the variation for each network parameter, which obeys the chain rule in deep neural network.

**Q3.13:** *At the beginning of section 4, please put the doi related content in the section of 'Data availability'. Additionally, please remove the duplicate information that is already mentioned in the Method. Please only present your results in the Result section.*

**Response:** Thanks for this comment. We have supplemented the doi related content at the beginning of section 4 as follow. Additionally, the duplicate information has been removed in section 4.

"It should be highlighted that this dataset can be directly downloaded at https://doi.org/10.5281/zenodo.4417458 for free use."

**Q3.14:** *Figure 10, the original patch shows almost the same as the reconstructed. Do you mean the original patch is missing here? I am sorry if I misunderstand.*

**Response:** Thanks for this question. In the simulated missing regions validation, six simulated square missing patches are performed in six continents based on the original soil moisture products (As the referee supposed that the original patch is missing). Through this way, we can easily compare the reconstructed SM regions with original SM regions, to validate the 2D spatial continuity of the proposed SGD-SM products. Detailed original and reconstructed spatial information of four simulated patches in 2015.7.25 are displayed in Fig. 10.

[Figure]

**Fig. 10.** Detailed original/reconstructed spatial information of four simulated patches in 2015.7.25

**Response:** Thanks for this issue. We have appended the black circles in Fig. 12(b) and (c), as shown below:

[Figure]

(a) Original

(b) Time-series averaging         (c) Proposed

**Figure 12.** Original/time-series averaging/proposed global soil moisture results in 2016.9.10

**Q3.16:** *Please describe uncertainties in this generated SGD-SM product.*

**Response:** Thanks for this significant comment. The uncertainties in this generated SGD-SM product can be classified as three types: 1) The errors of original AMSR2 SM product; 2) The meteorological factors such as precipitation and snowfall; 3) The generalization of proposed reconstructing model. Detailed descriptions of these three uncertainties are listed as follows:

1) The errors of original AMSR2 SM product: The proposed SGD-SM product is generated based on original AMSR2 SM product. While this original AMSR2 SM product also exists errors, due to the satellite sensor imaging and SM retrieval algorithm. As shown in Table 1, the R, RMSE, and MAE evaluation indexes of the original AMSR2 SM product are 0.687, 0.095, and 0.078, respectively. These errors are also inevitably transmitted into the generated SGD-SM product.

2) The meteorological factors: SGD-SM relies on the temporal continuity and spatial consistency for daily SM gap-filling. Nevertheless, if the unusual meteorologic occurs in single day such as precipitation and snowfall, it may destroy above assumption and influence the reconstructing effects. This uncertainty can be noticed in time-series validation, especially for rainy season.

3) The generalization of proposed reconstructing model: In this work, we train the proposed network through selecting complete soil moisture patches. In addition, the simulated masks are also chosen from the daily soil moisture products. However, it still exists the differences between the training data and testing data, such as land covering type, mask size, and so on. This uncertainty may disturb the generalization of proposed reconstructing model, to some degree.

**Table 1.** Comparisons between original and reconstructed soil moisture products

| Soil Moisture Productions | Evaluation index | | |
|:---:|:---:|:---:|:---:|
| | R | RMSE | MAE |
| Original | 0.687 | 0.095 | 0.078 |
| Reconstructed | 0.683 | 0.099 | 0.081 |

**Minor comments:**

**Q3.17:** *Please follow the "ESSD Manuscript composition (https://www.earth-system-science-data.net/submission.html/#manuscriptcomposition)" to make all related, e.g., Data availability as a separate section and use Sect accord with regulations.*

**Response:** Thanks for these suggestions. According to the manuscript composition, we have made all related parts (such as data availability, code availability, and author contributions) as the separate sections. The abbreviation 'Sect.' is also employed in our revised manuscript.

**Q3.18:** *Use 'besides' too many times in a scientific paper.*

**Response:** Thanks for this issue. We have rewritten the whole manuscript and removed most worthless 'besides' words.

**Q3.19:** *In line 20, I do not think ESA CCI is a sensor. Please revise.*

**Response:** Many thanks for pointing out this mistake! We have corrected this sentence and deleted 'ESA CCI is a sensor' in the revised version.

**Q3.20:** *'Products' not 'Production'*

**Response:** Thanks for this comment. We have replaced all the 'productions' with 'products' in our revised manuscript.

---

## Author Comment (AC4) · 12 Jan 2021

**Response to the Comments of Referee #4**

Dear Referee #4:

We are particularly grateful for your careful reading, and for giving us many constructive comments of this work!

According to the comments and suggestions, we have tried our best to improve the previous manuscript essd-2020-353 (SGD-SM: Generating Seamless Global Daily AMSR2 Soil Moisture Long-term Productions (2013–2019)). An item-by-item response follows.

Once again, we are particularly grateful for your careful reading and constructive comments. Thanks very much for your time.

Best regards,

Qiang Zhang

**General comments:**

*The authors present a method and dataset to fill daily AMSR2 soil moisture product gaps with a CNN for the years 2013-2019. The abstract jumps quickly into the topic, but is somehow ambiguous by not being clear to which current soil moisture products ("...the acquired daily soil moisture productions") the introduction relates? It would be good to explicitly state that it relates to the AMSR2 products (not productions) ...this lack of language clarity (e.g. "reliable cooperativity..." etc.) traces through the whole manuscript and needs to be strictly revised before considering acceptance. Otherwise, it is really hard for the reader to understand, and thus, to estimate the usefulness of this dataset. I think there is a lot of potential in this paper and dataset (even if "only" 6 years length). However, many technical aspects make it hard to grasp the content and estimate the quality in the first place. A few other comments as follows.*

**Response:** We are particularly grateful to the referee's careful reading and detailed suggestions!

"the acquired daily soil moisture productions" has been corrected as "the acquired daily AMSR2 soil moisture products" in the abstract part.

For the language clarity, we have revised the whole manuscript sentence by sentence in the updated version. According to the comments, we have tried our best to improve the previous manuscript. An item-by-item response to each constructive comment follows.

**Major comments:**

**Q4.1:** *Still the abstract creates more questions than answers. The evaluation measures are difficult to interpret. Why stating 2 evaluation measures, with one for original data? Also the choice of units ($m^3/m^3$) is not immediately clear as the dataset only produces percent values for soil moisture?*

**Response:** Thanks for these comments. For in-situ validation (2 evaluation measures), we compare the reconstructed with original AMSR2 daily soil moisture products as 'A (B)'. 'B' refers to the evaluation index of orginal products. 'A' stands for the evaluation index of reconstructed products after gap-filling. Compared with 'A' and 'B', the difference is that the reconstrcting values of gap regions need also to be evaluated in 'A', while 'B' needn't. Overall, the accuracy of reconstructed AMSR2 daily soil moisture products is generally accorded with the original products. The differences of these indexes R: 0.683 (0.687), RMSE: 0.099 (0.095), and MAE: 0.081 (0.078) are minor between the reconstructed and original soil moisture products. In other words, this validation ensures the reliability and availability of the proposed seamless global daily AMSR2 soil moisture products.

For the units of AMSR2 soil moisture products, we have corrected the "$m^3/m^3$" value as the percent value in the whole manuscript.

**Q4.2:** *The introduction is not clearly introducing the AMSR2 original dataset. I think it would be of great value if at least basic technical cornerstone of the original dataset is described.*

**Response:** Thanks for this meaningful suggestion. We have described the basic technical cornerstone of the original AMSR2 soil moisture products in sect 2.1 as follow:

"In this work, we focus on dealing with AMSR2 soil moisture products. This sensor was onboard on the Global Change Observation Mission 1-Water (GCOM-W1) satellite, launched in May 2012 (Kim et al., 2015). The released datasets include three passive microwave band frequencies: 6.9 GHz (C1 band), 7.3 GHz (C2 band, new frequency compared with AMSR-E), and 10.7 GHz (X band). It can observe the global land two times within a day (Wu et al., 2016): ascending (day-time) and descending (night-time) orbits. The primary spatial resolution of this datasets denotes $0.25°$ global grids. And the AMSR2 soil moisture retrieval algorithms include

Land Parameter Retrieval Model (LPRM) and Japan Aerospace Exploration Agency (JAXA) (Du et al., 2017; Kim et al., 2018). Besides, the uncertainty of soil moisture for each band were also given in AMSR2 products.

In our study, we choose LPRM AMSR2 descending level 3 (L3) global daily $0.25°$ soil moisture products as research data. This dataset could be obtained at https://hydro1.gesdisc.eosdis.nasa.gov/. For instance, the original AMSR2 $0.25°$ soil moisture data obtained in April 2, 2019 is displayed in Fig. 1. Due to the satellite orbit coverage and the limitations of soil moisture retrieving algorithms in tundra areas (Muzalevskiy et al., 2020), the acquired AMSR2 daily soil moisture products are always incomplete in global land (about $30\%\sim80\%$ invalid ratio, excluding Antarctica and most of Greenland), as shown in Fig. 1. The daily global land coverage ratio of AMSR2 soil moisture data in 2019 is listed in Fig. 2. Distinctly, the global land coverage ratio is low in wintertime, and high in summertime. The mean global coverage ratio of 2019 is just about $56.5\%$ in AMSR2 soil moisture daily products. Apparently, these incomplete soil moisture data cannot be directly applied for subsequent spatial and time-series analysis, as mentioned in previous Sect 1."

**Q4.3:** *Stating that the codes are also published is misleading so far. "The related Python codes of this dataset are also available at https://github.com/qzhang95/SGD-SM." (authors) only holds an example code of extracting data. I think it would be really helping the transparency of the data quality if the physical network implementation (TensorFlow, Keras, SKLearn, Pytorch?) would be also open in the spirit of open data and open source and reproducibility. Understandably, a trained neural network model is not 100% reproducible, but the model could be also archived on Zenodo? It only makes sense, because it would be very feasible to update the dataset on a yearly basis with the developed model. In the end the idea of ESSD is "living data".*

**Response:** Thanks for this issue. We have revised the sentence "The related Python codes of

this dataset are also available at https://github.com/qzhang95/SGD-SM." as "An example Python code of extracting this dataset are also available at https://github.com/qzhang95/SGD-SM." The training and testing procedure of the proposed model are implemented by Pytorch platform. The implemented code of this work will be released on Zenodo after possible acceptance, to flexibly update the dataset on a yearly basis with the developed model. We couldn't agree more with the referee's opinion that the idea of ESSD is "living data". This can also facilitate the development and utilization of soil moisture products.

**Q4.4:** *The year 2013 folder only contains 362 files, not 365. May 2013 only seems to have 28 files? Please check your upload on Zenodo.*

**Response:** Many thanks for your careful checking of the released dataset! The reason is that the NASA's official LPRM AMSR2 L3 soil moisture products indeed only have 28 daily files in May 2013 (missing data files in date May 11, 12, and 13). Referee can also verify this issue at NASA's GES DISC website at: https://hydro1.gesdisc.eosdis.nasa.gov/data/WAOB/LPRM_AMSR 2_D_SOILM3.001/2013/05/. We have supplemented this explanation in the updated products (https://doi.org/10.5281/zenodo.4417458).

**Q4.5:** *(Zhang et al., 2020) This citation is not in the references list, there are only Zhang 2020a and 2020b. Please add, presumably it is your data citation: DOI: 10.5281/zenodo.3960425*

**Response:** Thanks for this comment. We have added this citation into the reference of this work: Zhang, Q., Yuan, Q., Li, J., Wang, Y., Sun, F., Zhang, L. (2021). SGD-SM: Generating Seamless Global Daily AMSR2 Soil Moisture Long-term Products (2013-2019) (Version 1.0) [Data set]. Zenodo. DOI: 10.5281/zenodo.4417458.

**Q4.6:** *"More details of this work are released at https://qzhang95.github.io/Projects/Global-Daily-Seamless-AMSR2/..." your current paper should reflect the most important and up to date source of information until it is published. It would be ok though to refer to it as a "technical supplement" maybe? However, those URLs are not reliable. Thus, if you have a technical supplement with more details, it could be added to your Zenodo archive (which will be reliable and has a DOI).*

**Response:** Thanks for this comment. We have deleted this URL in the revised manuscript, to avoid misunderstanding this work. Some technical explanations with more details have been supplemented in our Zenodo archive (DOI: 10.5281/zenodo.4417458).

**Q4.7:** *Section 3.1 starting on page 7 is the methodological main part of your neural network implementation. While I'd like to acknowledge that technical level of description, I have two contradicting issues with it:*

*1) p7 ll137-139: "This network includes 11 layers (3D partial CNN unit and ReLU (Rectified Linear Unit)) in Fig. 4. The size of 3D filters is all set as 3×3×3. Number of feature maps before ten layers is fixed as 90, and the channel of feature map in the final layer is exported as 1".*

*That is very technical, yet it is not clear why those dimensions where chosen. The discussion section does not discuss the CNN and the design choices at all and what effect they have. For example, how can this capture the comparatively big gap areas of the original AMSR2 dataset?*

*2) On the other hand, much of this could also go into a technical supplement and you could provide a much higher-level overview for the reader in the paper. The paper is the data description, and many readers and future users of the dataset will not have the technical understanding of judging or even reading through the technical low-level design of the CNN – nevertheless this still also needs to be documented.*

**Response:** Thanks for these meaningful issues. We have added a technical supplement of the network implementation. Detailed effects of 3D filters, layers, and feature maps in the proposed model are depicted below:

1) 3D CNN filters: 3D CNN filters are employed to simultaneously capture both spatial and temporal soil moisture information in time-series products. For large gap areas, partial CNN is developed to exclude the invalid AMSR2 soil moisture information.

2) Layers: More layers in deep neural network can extract more intrinsic feature information for soil moisture products gap-filling.

3) Feature maps: Feature maps get the description of the original soil moisture products from multiple angles, through different 3D CNN filters.

For clearly understanding the parameters chosen in the proposed network such as 3D CNN filters, layer numbers, and feature maps, we have supplemented the sensitivity analysis of these parameters in discussion part. As listed in Table 4, Table 5, and Table 6, discussions for the 3D CNN filters, layer numbers, and feature maps are investigated in simulated missing regions validation, respectively. Accordingly, the optimal indexes are chosen as the setting value.

**Table 4**. Discussion for the 3D CNN filters in simulated missing regions validation

| Parameter | Evaluation index | | |
| --- | --- | --- | --- |
| | R | RMSE | MAE |
| 3×3×3 | **0.968** | **0.068** | **0.047** |
| 5×5×5 | 0.957 | 0.076 | 0.048 |
| 7×7×7 | 0.949 | 0.081 | 0.050 |

**Table 5**. Discussion for the layer numbers in simulated missing regions validation

| Parameter | Evaluation index | | |
| --- | --- | --- | --- |
| | R | RMSE | MAE |
| 10 | 0.962 | 0.072 | 0.048 |
| 11 | **0.968** | **0.068** | **0.047** |
| 12 | 0.966 | 0.070 | 0.049 |

**Table 6**. Discussion for the feature maps in simulated missing regions validation

| Parameter | Evaluation index | | |
|:---:|:---:|:---:|:---:|
| | R | RMSE | MAE |
| 60 | 0.963 | 0.071 | 0.048 |
| 90 | **0.968** | **0.068** | **0.047** |
| 120 | 0.967 | 0.069 | 0.047 |

**Q4.8:** *Also, why not an LSTM type network?*

**Response:** Thanks for this interesting query. LSTM type network indeed plays an important role for time-series data. In fact, the spatial and temporal information are both significant on spatial consistency between the valid and invalid soil moisture regions, and temporal continuity in time-series curve. Therefore, we develop the spatio-temporal convolutional network in this study, to simultaneously utilize the spatial and temporal soil moisture information. Through this 3-D strategy, we can both exploit the spatial consistency and temporal continuity for soil moisture products gap-filling. In our future work, we will combine the LSTM network with 3-D partial convolutional network, to futher ullize the spatio-temporal soil moisture information.

**Q4.9:** *Calling it spatio-temporal 3D might be misleading, as it is areal 2D and then a temporal dimension. Spatio-temporal indicates that already, the added 3D might lead to think of spatial 3D plus time.*

**Response:** Thanks for this comment. To avoid misleading, we have corrected "spatio-temporal 3D" as "spatio-temporal" in the whole manuscript.

**Q4.10:** *What was the reason to choose the 0.25 dec degrees as spacing for the data files?*

**Response:** Thanks for this query. The spatial resolution of the original global daily AMSR2 soil moisture products is 0.25 dec degrees. To avoid introducing additional error and uncertainty, we didn't carry out the downscaling operation of the generated SGD-SM products. We have supplemented this explanation in the revised manuscript.

**Q4.11:** *Last but not least, I'd like to advocate for a bit more metadata in the netcdf files, because netcdf provides great means for metadata. For example, you could adhere a bit more to the NetCDF-CF conventions, or at least add e.g. attributes such as title, reference and a time stamp in the dataset, not relying on the filename for example. You could also join at least the yearly slices into a "cube" that follows conventions of the Earth Sciences community (e.g. longitude instead of lon as variable name). Also the Zenodo deposit could have more fields filled out for improved discovery, more keywords (e.g. "soil moisture"?) and terms from controlled vocabularies, such as GEMET (https://www.eea.europa.eu/help/glossary/gemet-environmental-thesaurus ) or similar.*

**Response:** We are very grateful for referee's detailed guidance on our released dataset! We have regenerated the SGD-SM products and updated them on Zenodo flatform (DOI: 10.5281/zenodo.4417458). The title, reference, and time stamp have been added into the metadata in daily NetCDF files, as shown in the following table. In addition, we have also replaced the abbreviation variables "lon" and "lat" with the full names "longitude" and "latitude" in all the NetCDF files.

More keywords like soil moisture, AMSR2, seamless, global, daily, and SGD-SM have been supplemented in Zenodo flatform (https://doi.org/10.5281/zenodo.4417458), for better improving the utilization of our products in Figure A.

```
netcdf file: D:/SGD-SM/2019/LPRM_AMSR2_20190101.nc
{
  float Latitude(Latitude=720);
      :units = "degree_north";
  float Longitude(Longitude=1440);
      :units = "degree_east";

  global attributes:
      :reference = "SGD-SM: Generating Seamless Global Daily AMSR2
          Soil Moisture Long-term Products (2013-2019)";
      :url = "https://doi.org/10.5281/zenodo.3960425";
      :time_stamp = "2021-01-04 20:32:22";
      :author = "Processed by Qiang Zhang, Wuhan University";
      :date = "20190101";
      :source = "netCDF4 python module tutorial";
}
```

[Figure]

**Figure. A.** Updated keywords in Zenodo flatform (DOI: 10.5281/zenodo.4417458).

---

## Author Response (AR2)

**Author's Response for All the Comments**

Dear Topic Editor and Referees:

Once again, we are particularly grateful for your careful reading and for giving us many constructive comments of this work!

According to the second round of comments and suggestions, we have tried our best to improve the revised manuscript essd-2020-353 (Generating Seamless Global Daily AMSR2 Soil Moisture (SGD-SM) Long-term Products for the Years 2013-2019). The modified words or sentences are marked as blue color in the revised manuscript. An item-by-item response follows.

Thanks very much for your time.

Best regards,

Qiang Zhang and all co-authors

**General comments:**

*Great thanks to the authors for their efforts in addressing the comment. This reviewer has three major comments for the author' further consideration, and suggests to accept this paper after minor revisions.*

**Response:** We are particularly grateful to the reviewer for his/her careful reading and detailed suggestions once again! According to the reviewer's comments, we have tried our best to improve the previous revised manuscript. An item-by-item response to each major and minor constructive comment follows.

**Major comments:**

**Q3.1:** *Figure 1b shows no selected in-situ measurements at China and Russia for comparisons. As such, the reviewer is a bit less convinced by the quality of data at Asia, especially at the Tibetan Plateau where the fast changing atmospheric conditions complicate soil moisture retrieval and therefore the AMSR2 product. The author may check 'Zhang P, Zheng D, van der Velde R, Wen J, Zeng Y, Wang X, Wang Z, Chen J, Su Z. Status of the Tibetan Plateau observatory (Tibet-Obs) and a 10-year (2009–2019) surface soil moisture dataset. Earth System Science Data Discussions. 2020 Oct 16:1-34.' to see if these in-situ data can be applied for comparisons at this region. On the other hand, it seems that there are also in-situ soil moisture measurements at a national network of Chinese Automatic Soil Moisture Observation Stations (CASMOS) maintained by the Chinese Meteorological Administration (please see the information mentioned in https://hess.copernicus.org/preprints/hess-2020-407/), the author may also check the possibility of selecting some data to compensate comparisons. In case not lengthen the paper, comparisons can be attached as supplementary materials.*

**Response:** Thanks for this significant suggestion. Available in-situ measurements at China and Russia are rare for comparisons in ISMN. It is really a bit less convinced by the quality of data at Asia in previous validation. To overcome this issue, we have downloaded the Tibetan Plateau observatory (Tibet-Obs) soil moisture sites [1-2] between 2009 to 2019 at https://doi.org/10.4121/uuid:21220b23-ff36-4ca9-a08f-ccd53782e834. Then we selected the daily soil moisture sites (Maqu, Naqu, Ali and Shiquanhe networks, 0-5cm depth) in 2018 to validate the accuracy of our SGD-SM products in Tibet Plateau region. This surface SM dataset includes the original 15-min in situ measurements collected by multiple SM monitoring sites of the three networks. The comparison results are listed in the supplementary material. As shown in Table A, the reconstructed SGD-SM products also perform approximatively with the original AMSR2 products (Reconstructed (Original). R: 0.654 (0.657), RMSE: 0.097 (0.096), MAE: 0.083 (0.081).). This validation also demonstrates the availability of the proposed SGD-SM products at the Tibetan Plateau region, where exits the fast changing atmospheric conditions.

**Table A.** Comparisons between original and reconstructed soil moisture products in Tibetan Plateau via Tibetan Plateau observatory (Tibet-Obs) soil moisture sites.

| Soil Moisture products | Evaluation index | | |
|:---:|:---:|:---:|:---:|
| | R | RMSE | MAE |
| Original | 0.657 | 0.096 | 0.081 |
| Reconstructed | 0.654 | 0.097 | 0.083 |

References:

[1] Zhang, P., Zheng, D., van der Velde, R., Wen, J., Zeng, Y., Wang, X., Wang, Z., Chen, J., and Su, Z.: Status of the Tibetan Plateau observatory (Tibet-Obs) and a 10-year (2009–2019) surface soil moisture dataset, Earth Syst. Sci. Data Discuss., in review, 2020.

[2] Qiu, J., Dong, J., Crow, W. T., Zhang, X., Reichle, R. H., and M. De Lannoy, G. J.: The added value of brightness temperature assimilation for the SMAP Level-4 surface and root-zone soil moisture analysis over mainland China, Hydrol. Earth Syst. Sci. Discuss., in review, 2020.

**Q3.2:** *In Line 250, the author mentioned that 125 soil moisture stations (0-10cm) were used. Since AMSR2 are C-band (wavelength ~5 cm) sensors, meaning low penetration depth, even for SMAP L-band (wavelength ~21 cm), it can only detect changes of soil moisture in the top 0-5 cm (see F. T. Ulaby, R. K. Moore, and A. K. Fung, Microwave Remote Sensing, Active and Passive, Vol. III: From Theory to Applications. Norwood, MA: Artech House, 1986). The reviewer wonders why the author did not use 0-5 cm soil moisture measurements for comparisons. Unless the author provides the rationality, otherwise the review would suggest to use soil moisture measured at 0-5 cm for comparisons.*

**Response:** Thanks for this comment. As the reviewer stated, 0-5cm soil moisture values are more relevant with the AMSR2 soil moisture products. Therefore, we just employ the 0-5cm sites (113 stations after selecting) for comparisons. The updated in-situ validation between original and reconstructed soil moisture products by 0-5cm sites is listed in Table 1 as follow. Overall, the accuracy of reconstructed products is generally accorded with the original products. The differences of these indexes are minor between the original and reconstructed results in Table 1.

**Table 1.** Comparisons between original and reconstructed soil moisture products

| Soil Moisture products | Evaluation index | | |
| --- | --- | --- | --- |
| | R | RMSE | MAE |
| Original | 0.689 | 0.093 | 0.077 |
| Reconstructed | 0.685 | 0.097 | 0.079 |

**Q3.3:** *The review also suggests to make the code of the 3-D CCN network model available with the published paper.*

**Response:** Thanks for this suggestion. We have publicly released our SGD-SM model and codes

at https://github.com/qzhang95/SGD-SM (Language: Python 3.7.4; Flatform: Pytorch 1.7.1). This explanation has been supplemented in the updated version.

**Minor comments:**

**Q3.4:** *Please the author may complete the title as "Generating Seamless Global Daily AMSR2 Soil Moisture (SGD-SM) Long-term Products for the Years 2013-2019".*

**Response:** Thanks for this comment. We have completed the title as "Generating Seamless Global Daily AMSR2 Soil Moisture (SGD-SM) Long-term Products for the Years 2013-2019".

**Q3.5:** *In Line 19, please give the full name when it (i.e., EOS) is first mentioned.*

**Response:** Thanks for this issue. We have given the full name when it is first mentioned, such as Earth Observing System (EOS) in the whole manuscript.

**Q3.6:** *In Line 22, please add 'see Figure 1a.' for clarity in 'about 30%~80% missing ratio in AMSR2)'.*

**Response:** Many thanks for this helpful suggestion! We have added 'see Fig. 1(a)' for clarity in 'about 30%~80% missing ratio in AMSR2' in this sentence.

**Q3.7:** *In Line 143, please add 'detailed technique descriptions of the network implementation are provided in the supplementary materials' to guide the audience to see the supplementary.*

**Response:** Thanks for this comment. We have added 'Detailed technique descriptions of the network implementation are provided in the supplementary material', to guide the reader to read the supplementary file.

**Q3.8:** *In Line 221, please delete 'It should be highlighted that'.*

**Response:** Thanks for this issue. We have deleted 'It should be highlighted that...' in this sentence.

**Q3.9:** *Please explain 'COSMOS' in Figure 7 and give the WGS84 geographical latitude and longitude for these sites shown in the figure title.*

**Response:** Thanks for this comment. We have supplemented the explanation of 'COSMOS' in Figure 7 as 'COsmic-ray Soil Moisture Observing System (COSMOS)'. Besides, the WGS84 geographical latitude and longitude for these sites in Figure 7 are given in each sub-figure's title, respectively.

[Figure]

(a) C-026 (40.0062°N, 88.2904°W) (b) C-055 (0.2825°N, 36.8669°E) (c) C-098 (44.5711°N, 11.5321°E)

(d) C-044 (21.6176°S, 47.6325°W) (e) C-033 (37.0310°N, 119.2564°W) (f) C-076 (14.1592°S, 131.3881°E)

(g) C-087 (48.1411°N, 15.1702°E) (h) C-048 (48.3077°N, 105.1019°W) (i) C-012 (19.7650°N, 155.4234°W)

**Figure 7**. Scatters of the in-situ/reconstructed soil moisture values within selected COsmic-ray Soil Moisture Observing System (COSMOS) stations

**Author's Response to the Comments of Referee #4 (RC4)**

**General comments:**

> *The authors have significantly improved the manuscript and addressed the points raised by my comments. I am still wary about the comment, that they may only publish their processing scripts/Pytorch model AFTER possible acceptance. I guess it is a trust issue. However, the preprint is public under discussion, thus, the precedent of their work is claimed. It does not improve the credibility to expect that they might publish the codes or not. No reasoning is given as, why they would withhold the codes.*
>
> *Nonetheless, while a lot of text was added and/or edited, the overall language quality still needs to be improved, e.g. such "The error of soil moisture for each frequency were also given in AMSR2 products." -> the errors were given | the error was given ... etc.*

**Response:** We are particularly grateful to the reviewer for his/her careful reading and detailed suggestions! According to the comments, we have tried our best to improve the previous revised manuscript.

For the codes issue of this work, we have publicly released our SGD-SM model and codes at https://github.com/qzhang95/SGD-SM (Language: Python 3.7.4; Flatform: Pytorch 1.7.1). This explanation has been supplemented in the updated version.

For the language quality, we have revised the whole manuscript sentence-by-sentence. Some grammar mistakes have also been corrected in this work, such as "the error were given..." to "the error was given".